# SecA2 Associates with Translating Ribosomes and Contributes to the Secretion of Potent IFN-β Inducing RNAs

**DOI:** 10.3390/ijms232315021

**Published:** 2022-11-30

**Authors:** Lisa Teubner, Renate Frantz, Luigi La Pietra, Martina Hudel, Jasmin Bazant, Günter Lochnit, Lena Eismann, Günter Kramer, Trinad Chakraborty, Mobarak Abu Mraheil

**Affiliations:** 1Institute of Medical Microbiology, German Center for Infection Giessen-Marburg-Langen Site, Justus-Liebig University Giessen, Schubertstrasse 81, 35392 Giessen, Germany; 2Protein Analytics, Institute of Biochemistry, Justus-Liebig University Giessen, 35390 Giessen, Germany; 3Center for Molecular Biology of Heidelberg University (ZMBH), German Cancer Research Center (DKFZ), DKFZ-ZMBH Alliance, Im Neuenheimer Feld 282, 69120 Heidelberg, Germany

**Keywords:** *Listeria monocytogenes*, SecA2, secreted RNA, IFN-beta

## Abstract

Protein secretion plays a central role in modulating interactions of the human pathogen *Listeria monocytogenes* with its environment. Recently, secretion of RNA has emerged as an important strategy used by the pathogen to manipulate the host cell response to its advantage. In general, the Sec-dependent translocation pathway is a major route for protein secretion in *L. monocytogenes*, but mechanistic insights into the secretion of RNA by these pathways are lacking. Apart from the classical SecA1 secretion pathway, *L. monocytogenes* also encodes for a SecA paralogue (SecA2) which targets the export of a specific subset of proteins, some of which are involved in virulence. Here, we demonstrated that SecA2 co-sediments with translating ribosomes and provided evidence that it associates with a subset of secreted small non-coding RNAs (sRNAs) that induce high levels of IFN-β response in host cells. We found that enolase, which is translocated by a SecA2-dependent mechanism, binds to several sRNAs, suggesting a pathway by which sRNAs are targeted to the supernatant of *L. monocytogenes*.

## 1. Introduction

Successful propagation of intracellular bacteria requires modulation of host cellular responses. To achieve this, bacteria secrete a large number of effector proteins to trigger for entry into host cells, to escape cellular defense mechanisms, and to moderate host antipathogen responses. The facultative intracellular pathogen *Listeria monocytogenes* possesses six protein secretion systems: (1) the Secretion pathway (Sec); (2) the twin-arginine translocation pathway (Tat); (3) the flagellum exporter apparatus (FEA); (4) the fimbrillin protein exporter (FPE); (5) Holin; and the (6) WXG100 system [1,2,3,4]. The Sec system is the major pathway by which listerial proteins are secreted. It is composed of the membrane-located heteromeric SecYEG translocon and the SecA ATPase, which provides the necessitated energy for the translocation [5]. The Sec system is essential for the translocation of proteins that are important for the biogenesis of cell walls, membranes, motility, nutrient uptake, and proteins that are associated with pathogenesis and symbiosis [6,7]. SecA (referred to here as SecA1) is an essential housekeeping cytosolic ATPase composed of multiple domains that undergo a series of conformational changes fueled by cycles of ATP binding and hydrolysis that are associated with translocation of the target protein through the SecYEG membrane channel [8]. A subset of pathogenic Gram-positive bacteria, including *L. monocytogenes*, possess an additional SecA protein, namely, SecA2 [9,10]. Based on the substrates and interacting partners of SecA2, it has been proposed that there are two mechanistically distinct types of SecA2 pathways: the accessory SecA2/SecY2 system, which transports a single specific substrate; and the multi-substrate SecA2 system, which occurs in *L. monocytogenes* and uses the SecYEG channel instead of SecY2 [11]. SecA2 is very similar in sequence, structure, and function to SecA1, but it exhibits a truncation in the C-terminal linker region (CTL) [11]. Unlike SecA1, the SecA2 pathway is often dispensable for bacterial growth, with the exception of *Corynebacterium glutamicum* [12] and *Clostridium difficile* [13]. Several studies have examined the contribution of SecA2 to *Listeria* virulence [14,15]. *L. monocytogenes* secretes 19 proteins in a SecA2-dependent manner [4,16]. The cell wall-secreted hydrolases autolysin p60 (protein of 60 kDa, or Iap (Invasion associated protein) also called CwhA (cell wall hydrolase A)) and the muramidase MurA (also called NamA [N-acetylmuramidase A]), known to be involved in cell division, are well known SecA2 substrates [16,17,18]. *Listeria* lacking SecA2 show reduced secretion of both p60 and MurA in the supernatant of growing bacteria, and exhibit a chain-like cell morphology manifesting in a rough colony type [15,17]. The reduced secretion of SecA2-dependent proteins such as p60 has been implicated in the diminished virulence phenotype [16,18,19,20]. Additionally, SecA2-inactivation affects significantly the degree of hydrophilicity and Lewis acid-based properties, and thereby affects the cell surface properties of *L. monocytogenes* [21]. *L. monocytogenes* lacking SecA2 were also found to be unable to generate long-term protective immunity in mice [20], to be poor inducers of type I interferon (IFN) response and to secrete lower amount of RNAs in the supernatant [22]. We showed that specific secreted small non-coding RNAs (sRNAs) of *L. monocytogenes* are potent activators of the type I IFN response [23]. Sensing of pathogen-derived RNA released from bacteria growing in the cytosol of infected host cells is increasingly recognized as a central mechanism for modulating innate immune activation during infection. In contrast to its role in defense against viral infections, induction of the type I IFN response is detrimental for the host during infection with intracellular bacteria such as *L. monocytogenes*, *Mycobacterium tuberculosis* and *Francisella tularensis* [23,24]. These findings underpin the significance of released RNA for *Listeria* as a strategy to modulate host responses to its advantage.

In this study, we found that SecA2 associates and co-sediments with translating ribosomes. By examining for those proteins that associate with purified SecA2, we identified 11 of 19 previously described SecA2-dependent secreted proteins [4,16]. Identification of SecA2-associated RNAs revealed the presence of small non-coding RNAs that induce high levels of interferon-β (IFN-β). Specifically, we found that enolase, a SecA2-dependent secreted protein, associates with a sRNA (rli32) that we previously identified as a strong and specific inducer of the type I IFN response [23]. These results identify a proxy mechanism by which SecA2 uses an RNA-binding protein substrate to translocate specific RNAs from cytoplasm to an environment external to the bacterium.

## 2. Results

### 2.1. Production of Tagged and Functional SecA2 in ∆secA2 Deletion Mutant

To produce a tagged SecA2 we fused the 5′ end of *secA2* coding sequence of *L. monocytogenes* EGD-e to a sequence encoding Strep-tag and a linker composed of two alanine residues. The Strep-tag-*secA2*-fusion product was inserted into the pERL-3 vector [25] downstream of the artificial highly expressed listerial promoter (P_help_) [26]. The *secA*2 containing pERL-3 vector was transformed in the ∆*secA2* deletion mutant (Lm-∆*secA2*) to generate the complemented strain Lm-∆*secA*2::SecA2_strep_. To assess the functionality of the tagged SecA2, we checked for two well-known phenotypes of the SecA2 deletion mutant in the complemented strain. First, we examined wild type Lm, Lm-∆*secA2* and Lm-∆*secA*2::SecA2_strep_ for changes in morphology during exponential growth phase using light microscopy. Lm-∆*secA2* form long cell chains due to a septal defect that results from lack of the secretion of p60/Iap and MurA/NamA [16,17,18] (Figure 1A). The complemented recombinant strain Lm-∆*secA*2::SecA2_strep_ had reverted to the single cell morphology seen with wild type (Lm) bacteria (Figure 1A). Second, we compared the consistency of the pellets of the three strains. The pellet of centrifuged Lm-∆*secA2* culture was loosely packed and was detachable by shaking the tube, while the pellet of Lm-∆*secA*2::SecA2_strep_ was consistent and resembled that of the wild type strain (Figure 1B). The recovered phenotypes of the complemented strain indicated the functionality of the generated SecA2.

### 2.2. Affinity-Based Purification of Tagged SecA2

Cytosolic protein extracts were prepared from exponentially grown Lm-∆*secA*2::SecA2_strep_ cultures for affinity-based purification of SecA2. Bacterial lysates were visualized by coomassie brilliant blue-stained SDS-PAGE gel before (Appendix A, BC = before column) and after passage through the affinity chromatography column (Appendix A, AC = after column). After eight wash steps (Appendix A), potential binding partners and associated substrates to SecA2 were eluted from the column using a desthiobiotin-containing buffer (Appendix A). The presence of SecA2 in BC, AC and elution fractions was verified using specific anti-SecA2 antibody (Figure 2A,B). To have a control for the subsequent experiments, we also generated and purified a Strep-tagged SecA1 of *L. monocytogenes* by using the same procedure applied for the production and purification of SecA2 (see above). The isolated SecA1 protein has a molecular weight of 96 kDa as compared to 90 kDa SecA2 counterpart (Figure 2C).

### 2.3. SecA2 Co-Sediments with Ribosomes

*Escherichia coli* SecA1 was previously shown to bind translating ribosomes [27,28]. This property has not yet been shown for any of the known SecA2 proteins described in Gram-positive bacteria. However, SecA1 and SecA2 of *L. monocytogenes* have an overall similarity of 46% at the amino acid level [11]. To examine the ability of the recombinant SecA2 to bind ribosomes, we performed co-sedimentation experiments using purified *L. monocytogenes* ribosomes and SecA2. Co-sedimentation experiments using purified SecA1 were performed in parallel as a control. Purified SecA2 and SecA1 proteins (1 µM) were mixed with purified ribosomes (4 µM). Subsequently, ribosome-bound SecA2 and SecA1 were separated from unbound protein by pelleting the high molecular weight ribosome complexes through a 20% sucrose cushion using ultracentrifugation. Figure 3A depicts the proteins of ribosomal pellet and Figure 3B demonstrates the corresponding supernatant. A Streptactin-HRP conjugate against the Strep-tag fused to SecA2 and SecA1 proteins was used to confirm that both proteins co-sediment with ribosomes to the pellet fraction (Figure 3C). However, unlike SecA1, a fraction of SecA2 remained in the supernatant post-centrifugation suggesting SecA2 binds ribosomes with reduced affinity as compared to SecA1 (Figure 3D). The efficient sedimentation of ribosomes is demonstrated by the exclusive detection of the ribosomal protein RplB in the ribosomal pellet but not the supernatant (Figure 3E,F). The requirement of the presence of ribosomes for SecA2 sedimentation is shown by the absence of SecA2 in the pellet (P) when the ribosomes are not added (Figure 3G). This control demonstrates that SecA2 sedimentation is not due to the pelleting of protein aggregates.

### 2.4. SecA2 Interacts with Translating Ribosomes

To establish that SecA2 specifically interacts with translating ribosomes, we prepared lysates from exponentially growing complemented strain and separated the 30S and 50S ribosomal subunits from translating ribosomes that sediment in the polysome fractions using sucrose gradient ultracentrifugation. The gradient was separated into 36 fractions, which were individually analyzed for SecA2 presence. SecA2 was detectable in the first six ribosome-free fractions, in all ribosomal fractions (30S, 50S, 70S) and in polysome fractions (Figure 4A,B). As expected, ribosomal RplB protein was detectable only in fractions 11 to 28 enclosing the large subunit, including polysomes (Figure 4C). This finding suggests that SecA2 interacts with idle and translating ribosomes. Similar results were also obtained using SecA1, which was previously shown to interact with ribosomes (Appendix A) [27]. The detected SecA2 amounts in the ribosome-free fractions (Figure 4B, first six fractions) are higher than those detected for SecA1 (Appendix A, first six fractions). These results correlate with the findings described above and indicate that SecA2 binds ribosomes, yet less efficiently than SecA1 (Figure 3D). Next, we examined if the ribosome profiles might be affected by the expression of the erythromycin resistance gene encoded on the plasmids, by comparing sucrose gradient fractions derived from wild type *L. monocytogenes* (Appendix A) and cells transformed with the empty (without *secA2*) vector pERL-3 (Appendix A). The results show no differences in the ribosome profiles by the expression of the erythromycin resistance gene.

### 2.5. Identification of SecA2 Co-Eluted Proteins

Using tandem mass spectrometry, we aimed to identify proteins that were co-purified with SecA2 by affinity chromatography. The elution fractions E1 to E6, in which the presence of SecA2 was confirmed via specific antibody (Figure 2B), were pooled and the co-eluted proteins were detected using tandem mass spectrometry. A total of 85 proteins were identified in SecA2 protein elutes (Table 1). Of the 19 proteins previously described to be secreted via SecA2 [16] we detected a total of 11 proteins (highlighted in blue) that were co-isolated with SecA2 from the cytosolic fraction (Appendix A). In a previous study of Lenz and coworker [16], the bacteria were grown in LB medium (not in BHI as done in our study) and the 19 proteins were detected by separation of supernatant and cell wall proteins on SDS-PAGE. The authors compared the protein patterns of wild type and Lm-∆*secA*2 deletion mutant. The different protein bands were then excised from the gel and identified. In this study, we identified proteins that were co-purified with SecA2 (in solution) by affinity chromatography. Therefore, we speculate that the combination of different growth media and preparation methods could be reasons for the differences in the detected proteins between the two studies. The detected proteins included the Hsp70 chaperone DnaK (DnaK), enolase (Eno), elongation factor Tu (Tuf) and 60 kDa chaperonin (GroE). Essential components for translation and aminoacyl-tRNA turnover, such as Tuf, the elongation factor G (Fus) and several aminoacyl-tRNA ligases (AlaS, IleS and GltX; Table 1), were found to be SecA2-associated. Both Tuf and the elongation factor Ts (Tsf) have been experimentally determined to be preset in the membrane, cell surface fractions and in extracellular milieu [4]. Several ribosomal proteins of *Listeria* were also among SecA2 co-isolated proteins. Two of them were described in a previous study to be extracellular located (RpsF and RplD), and additional ribosomal proteins such as RplT, RplB and RplO were found in the membrane fraction [4]. In addition, many RNA-binding proteins such as the ATP-dependent RNA helicase CshA (CshA), enolase (Eno), polyribonucleotide nucleotidyltransferase (PnpA), DNA-binding protein HU (Hup), the single strand nucleic binding cold shock-like proteins, CspLA (CspL) and cold shock-like protein CspLB (CspB) were found to co-isolate with SecA2. Apart from Eno, five of the ten glycolytic enzymes were co-isolated with SecA2: glucose-6-phosphate-isomerase (Pgi), phosphofructokinase-1 (Pfk), triose phosphate isomerase (Tpi), glyceraldehyde-3-phosphate dehydrogenase (GAPD) and phosphoglycerate kinase (Pgk) (Table 1). Several enzymes of the glycolytic pathway have been found to act as moonlighting proteins [29]. Additionally, stress-related proteins such as catalase (Cat), the surface proteins OppA and TcsA were also co-isolated with SecA2 (Table 1). Overall, the presence of proteins with functions in translation, protein modification, glycolysis and the presence of RNA degradosome-associated proteins strongly suggests a role for SecA2 in an interaction network comprising of the ribosome, degradosome, and components of central carbon metabolism pathways.

### 2.6. Association of RNA with SecA2

Deletion of *secA2* in *L. monocytogenes* reduces the amount of extracellular RNA [22]. An examination of RNA quantity in the supernatant of the complemented strain Lm-∆*secA*2::SecA2_strep_ revealed the presence of RNA in amounts that are higher than wild type (Lm) (Figure 5A). This is due to the increased expression of SecA2 by the highly expressed listerial promoter (P_help_) in the complemented strain (Appendix A). The levels of RNA detected in the supernatants of the Lm-Δ*secA*2 mutant was 2.5 fold lower than Lm-∆*secA*2::SecA2_strep_ (Figure 5A). To examine for SecA2-associated RNAs, we isolated RNA from wash, elution, BC, and AC fractions (Figure 5B). The results show a continuous decrease in RNA concentration between the first (W1) and the last wash step (W8) (Figure 5B). RNA isolation from the specific SecA2 eluted fractions (E1 to E9) showed clear correlation between SecA2 amounts and RNA concentration in the respective fraction (Figure 5B,C). Neither proteins nor RNAs were eluted from wild type cytosolic lysates, which lacks Strep-tagged SecA2 (Appendix A).

### 2.7. SecA2-Associated RNAs Are Potent Inducers of IFN-β

To assess the IFN-β inducing capability of SecA2-associated RNA, we transfected bone marrow-derived macrophages (BMDM). Control transfections were performed using cytosolic RNA fraction (Lm-cytosolic-RNA) together with secreted RNA fraction isolated from the supernatant of wild type cultures (Lm-sec-RNA), SecA2 deletion mutant (Lm-∆*secA2*) and the complemented strain (Lm-∆*secA2*::SecA2_strep_). Additionally, SecA2- and SecA1-associated RNAs were transfected in BMDM (Figure 6A). Transfection of 100 ng sec-RNA of wild type and the complemented strain showed significantly higher IFN-β induction as compared to Lm-∆*secA2*. SecA2-associated RNAs induced significantly higher IFN-β expression in macrophages when compared to SecA1-associated RNAs (Figure 6A). The highest IFN-β expression induction was measured in wild type sec-RNA transfected macrophages, whereas the IFN-β response triggered by cytosolic RNA of wild type was the lowest among all tested RNA fractions (Figure 6A). The profiles of SecA1- and SecA2-associated RNA revealed significant differences in RNA size distribution of both fractions. SecA2-associated RNAs represent mainly short RNA of 20 to 300 nucleotides (nt) (Figure 6B). In contrast, the size of SecA1-associated RNAs is more diverse, ranging from 20 to over 3000 nt (Figure 6C). The treatment of SecA2- and SecA1-associated RNAs with RNase led to a complete loss of the RNAs in the sample (Appendix A).

### 2.8. Detection of Small Non-Coding RNAs in SecA2-Associated RNAs

Next, RNA-Seq was performed to identify the RNA species associated with SecA2. The results showed that all types of RNA, including sRNA, mRNA, tRNA and rRNA occur in SecA2-associated RNA fraction (Appendix A). Interestingly, the data revealed a high proportion of sRNA in SecA2-associated RNAs (Appendix A). The highest read numbers of SecA2-associated sRNAs were found for the highly transcribed cytosolic sRNAs ssrA, the 4.5S RNA of SRP and ssrS (Figure 6D). Overall, four sRNAs (rli31, rli32, rli33-1 and rli47) of the detected SecA2-associated sRNAs (Figure 6D) are highly induced in *L. monocytogenes* during infection [31] of which rli32 is a very strong inducer of IFN-β [23].

### 2.9. Deletion of Enolase Affects the Sec-RNA Amount and Composition in the Sec-RNA Fraction

The mass spectrometry (MS) data demonstrated that enolase, a glycolytic enzyme with important functions in RNA turnover, was found to be one of the SecA2 co-isolated proteins (Table 1). Enolase is also known to be secreted in a SecA2-dependent manner [16] and has the propensity to bind RNA [32,33,34]. Therefore, we generated an enolase deletion mutant (Lm-Δ*eno*) and examined whether the absence of enolase leads to changes in the secreted RNA fraction. The total amount of secreted RNA isolated from Lm-Δ*eno* supernatant was significantly lower than that isolated from Lm wild type and comparable to the sec-RNA amounts obtained from Lm-Δ*secA2* (Figure 7A). The differences in the amounts of secreted RNA are not due to variations in bacterial growth because both deletion mutants show similar growth like wild type (Appendix A). The ability of sec-RNA isolated from Lm-Δ*eno* supernatant to induce IFN-β was significantly lower than the sec-RNA of Lm wild type and comparable to the IFN-β induction level of Lm-Δ*secA*2 (Figure 7B). No significant differences in the length distribution of sec-RNA between Lm wild type, Lm-Δ*secA*2 and Lm-Δ*eno* were detectable (Figure 7C–E). This suggested differences in the composition of sec-RNA. We therefore performed quantitative real-time PCR to determine the amounts of selected sRNA, known to be secreted [23], in the sec-RNA fractions of Lm-Δ*eno* and Lm wild type. The results revealed significant decrease in the amounts of four sRNAs (LhrA, SRP, rli32 and rli99) in the supernatant of Lm-Δ*eno* (Appendix A). Other sRNAs such as ssrS, rli47, rli51, and 16S rRNA are present in higher quantities in Lm-Δ*eno* (Appendix A). The quantitative-RT PCR results are given as Ct-values, as all validated reference genes for cytosolic RNA of Listeria [35] occur in different abundances in sec-RNA as shown e.g., for the reference gene 16S rRNA (Appendix A).

## 3. Discussion

SecA2 was first recognized twenty years ago in mycobacteria [14]. Although nonessential for *L. monocytogenes*, SecA2 is required for secretion of a specific subset of listerial proteins including the two abundant autolysins N-acetylmuramidase (NamA) and a secreted endopeptidase (p60) [16,17]. Impeded secretion of NamA and p60 leads to the formation of elongated bacterial filament-like chains, a phenotype characteristic for the ∆*secA2* deletion mutant [17]. SecA2 is also linked to the secretion of RNA because *secA2* deletion leads to an overall reduction in the amount of extracellular RNA in the SecA2-mutant strain [22]. Recently, SecA2 in *Mycobacterium tuberculosis* was found to be involved in RNA secretion, but the specific substrates involved in this process are yet unknown [36]. During *Listeria* infection secreted bacterial RNAs (sRNA) induce a type-I IFN response [22] which is beneficial for the intracellular replication of this pathogen [37,38,39]. More recently, a specific group of secreted sRNAs were identified as potent inducers of a RIG-I-dependent IFN-β response [23]. Another study discovered a secreted RNA-binding protein, termed Zea, which was found to be associated with a subset of *L. monocytogenes* RNAs, causing their accumulation in the extracellular medium. Zea binding to RIG-I of the host cell during *L. monocytogenes* infection enhances IFN-β production [40].

In the present study, we aimed at understanding the role of the SecA2 machinery in the secretion of nucleic acids. We used Strep-tag-versions of proteins because of the small size of the tag and neutral isoelectric point favors purification of protein-protein complexes under physiological conditions [41,42]. We first demonstrated that the tagged SecA2 version is functional and reverses the filamentous phenotype of the *secA*2 deletion mutant in the complemented strain. We provide information on SecA2 association with idle ribosomes as well as with translating polysomes. Although *E. coli* SecA1 was previously shown to bind translating ribosomes [27,28], SecA1 binding to translating ribosomes in *Listeria* has not been reported yet. Thus SecA2, like its SecA1 counterpart, may shuttle between plasma membrane located SecYEG and the cytosol to interact with translating ribosomes and nascent proteins [27]. SecA2 binding to translating ribosomes is further supported by the co-elution of many proteins that are implicated with translation such as RpsF, RplD RplT, RplB and RplO. Further SecA2 associated proteins included CshA, Eno and PnpA (polyribonucleotide nucleotidyltransferase), all of which are part of the RNA multiprotein macromolecular degradosome complex and have a crucial role in RNA turnover and maturation [43,44,45,46,47].

Six of the ten glycolytic enzymes viz. Eno, Pgi, Pfk, Tpi, GAPD and Pgk were detected in SecA2-associated protein fractions (Table 1). Eno, Tpi and GAPD have been found to possess moonlighting activities [29]. Streptococci and Staphylococci have most of the enzymes of the glycolytic pathway on their cell surfaces [48,49]. Excepting Tpi, all the other glycolytic enzymes have previously been experimentally determined to be either in the membrane or supernatant fractions [4]. The glycolytic enzyme enolase is a part of the RNA degradosome and has important functions in RNA turnover. Because enolase was (i) found to be one of the SecA2 co-isolated proteins in our MS data (Table 1), (ii) known to be secreted in a SecA2-dependent manner [16] and (iii) has the propensity to bind RNA [32,33,34], we generated a Lm-Δ*eno* deletion mutant to explore the role of enolase in RNA secretion. The total amount of sec-RNA isolated from Lm-Δ*eno* supernatant was significantly lower than sec-RNA isolated from wild type *L. monocytogenes* (Figure 7A) and comparable to that obtained from Lm-Δ*secA2*. The ability of sec-RNA isolated from Lm-Δ*eno* supernatant to induce IFN-β was significantly impaired when compared to sec-RNA obtained from wild type even though no significant differences in the profile of sec-RNA between wild type *L. monocytogenes* and Lm-Δ*eno* were detectable. As this finding indicates differences in the composition of sec-RNA, we determined the amounts of a set of secreted sRNAs [23] in the sec-RNA fraction of Lm-Δ*eno* and wild type *L. monocytogenes*. The results revealed significant decreases in the amounts of three sRNAs (rli32, rli99 and LhrA) in the supernatant of Lm-Δ*eno*. Interestingly, rli32 and rli99 were found to be potent inducers of the IFN-β response [23]. As it is known that enolase protects RNA from degradation, the decreased amounts of rli32 and rli99 detected in the isogenic enolase mutant probably account for the reduced IFN-β induction by the sec-RNA of Lm-Δ*eno*. Rli32 and rli99 do not belong to the Zea-associated sRNAs [40], which suggests that *L. monocytogenes* possess distinct pathways for RNA secretion.

Stress-related proteins such as catalase, and the surface proteins OppA and TcsA were also co-isolated with SecA2. These proteins were shown to be regulated by the small non-coding RNA LhrC in the context of heme toxicity [50]. Overall, the presence of proteins with functions in translation, protein modification, glycolysis and the presence of RNA degradosome-associated proteins strongly suggests a role for SecA2 in an interaction network comprising of the ribosome, the degradosome and components of central carbon metabolism. Interestingly, the extraction of SecA2-associated RNAs revealed unique enrichment of short RNAs yielding a profile that is highly similar to that of the recently described total sec-RNA [23]. Indeed, we found that SecA2-associated RNAs are significantly more potent in the induction of IFN-β than SecA1-associated RNAs. This is related to the presence of sRNAs such as rli31, rli33-1, rli32 and rli47 that are intracellularly expressed and highly enriched in the SecA2-associated RNA fraction [31,51]. On the other hand, several sRNAs found to be highly expressed in the cytosol of *L. monocytogenes* such as rli27, rli94, rli112 and rli50 [31] are absent or moderately present in SecA2-associated RNAs. Noteworthy, rli31, rli32, rli33-1 and rli47 belong to those sRNAs that are highly upregulated during intracellular growth of *L. monocytogenes* [31].

The results presented herein suggest a mechanism for the puzzling observation of RNA secretion by growing bacteria (Figure 8). We find that SecA2-dependent secretion of a subset of sRNAs is co-dependent on the various protein-substrates secreted by this translocation pathway. More specifically, we find that enolase, an RNA-binding protein, which has been previously demonstrated to be secreted by many bacteria, is involved in RNA secretion to the supernatant. The nature of this specific interaction remains to be elucidated but has major implications for the functionality of specific RNAs and their properties to interact with specific components of the innate immune response.

## 4. Materials and Methods

### 4.1. Bacterial Strains and Growth Conditions

Overnight cultures of *L. monocytogenes* EGD-e 1/2a were either grown in brain heart infusion (BHI) broth or in chemically defined medium [52] at 37 °C to an OD_600nm_ = 1.0. *Escherichia coli* DH10β cells (Invitrogen, Wesel, Germany) were cultured in Luria Bertani (LB) medium. For selection of *E. coli* harboring the plasmid pERL-3 [25], bacteria were grown in the presence of 300 µg/mL erythromycin. For *Listeria* cells harboring pERL-3 plasmid 5 µg/mL erythromycin was added to culture unless otherwise stated.

### 4.2. Construction of Deletion Mutants

The *secA2* deletion mutant (Lm-∆*secA2*) used was previously described in (17). The chromosomal deletion mutant of enolase (eno) was constructed by generating the 5′ (with primers “lmo2455_4f” and “lmo2455_3r”) and the 3′ (with primers “lmo2455_2f” and “lmo2455_1r”) flanking regions of *eno*. The primer sequences used to generate the isogenic mutants are presented in Appendix A. PCR fragments were purified and ligated into the temperature-sensitive suicide vector pAUL-A. Plasmid DNA of pAUL-A bearing the fragments were transformed in *L. monocytogenes* to generate the chromosomal deletion mutants as described previously [53].

### 4.3. Construction of Affinity-Tagged SecA2 Expression Plasmid

The 5′-end of *secA2* gene sequence of *L. monocytogenes* EGD-e 1/2a was fused with Strep tag WSHPQFEK (IBA, Göttingen, Germany) and the terminal sequence of hemolysin (*hly*) was fused to the 3′-end. The construct was synthesized by a commercial vendor (GeneArt AG, Regensburg, Germany). The plasmid pMK-RQ, harboring the recombinant *secA2* gene sequence, was digested with the restriction enzymes *Sac*I and *Xho*I and the resulting fragment was ligated in pERL-3 vector [25]. Competent bacteria cells of the *secA2* deletion mutant (Lm-∆*secA2*) were transformed with the plasmid to produce the recombinant Strep-tagged SecA2 protein. Transformed clones were selected on BHI agar plates containing 5 µg/mL erythromycin.

For construction of recombinant SecA1 the Strep tag sequence of the vector harboring recombinant *secA2* was amplified. The *secA1* gene sequence of *L. monocytogenes* EGD-e 1/2a was fused to a Strep tag encoding sequence and the *hly* terminal sequence and ligated in pERL-3 as described for *secA2*.

### 4.4. Protein Purification

All purification procedures were done at 4 °C to avoid protein degradation. Protein concentrations were estimated by Bradford assay (Bio-Rad, Hercules, CA, USA). Cells grown in BHI were harvested at OD_600nm_ = 1.0 by centrifugation at 16,000× *g*. The resulting cell pellet was washed and resuspended in 1× PBS (Dulbecco) and protease inhibitor cocktail (Biotool, Heidelberg, Germany) was added. Subsequently, bacterial cells were disrupted using Lysing Matrix B tubes and the FastPrep-24 instrument (MP Biomedicals, Eschwege, Germany). Cell lysates were centrifuged at 16,000× *g* for 30 min and the supernatant, which contained the cytosolic fraction, was sterile filtered using 0.22 µm filter (Merck, Darmstadt, Germany). The cytosolic fraction was used for SecA2 or SecA1 protein purification via affinity chromatography using StrepTactin^®^ Superflow^®^ high-capacity column with 1 mL bed volume (IBA lifesciences, Göttingen, Germany). The affinity chromatography was done according to manufacturer’s instructions, with the exception that washing steps were expanded to eight and the elution was repeated ten times. The composition of the buffers used in the purification: Wash buffer: 100 mM Tris, 150 mM NaCl, pH 8.0; Elution buffer: 100 mM Tris-Cl, 150 mM NaCl, 5 mM desthiobiotin, pH 8.0.

### 4.5. Matrix-Assisted Laser-Desorption Ionization Time-Of-Flight Mass Spectrometry (MALDI-TOF-MS)

Protein solutions were digested with trypsin and then desalted and concentrated. MALDI-TOF-MS was performed on an Ultraflex TOF/TOF mass spectrometer (Bruker Daltonics, Bremen, Germany) equipped with a nitrogen laser and a LIFT-MS/MS facility. The instrument was operated in the positive-ion reflectron mode using 2.5-dihydroxybenzoic acid and methylendiphosphonic acid as matrix. Sum spectra consisting of 200–400 single spectra were acquired. For data processing and instrument control the Compass 1.4 software package consisting of FlexControl 4.4, FlexAnalysis 3.4 4, Sequence Editor and BioTools 3.2 was used. External calibration was performed with a peptide standard (Bruker Daltonics).

Database search. Proteins were identified by MASCOT peptide mass fingerprint search (http://www.matrixscience.com) using the Swissprot database (2019_11, 561,568 sequence entries). For the search, a mass tolerance of 75 ppm was allowed, and oxidation of methionine as a variable modification was used. The search was restricted to bacteria. The database of Clusters of Orthologous Groups of Proteins (COGs) in Table 1 was used for the assignment of the proteins functions [54]. The Exponentially Modified Protein Abundance Index (emPAI) was used to estimate the absolute protein contents by the number of sequenced peptides per protein [55]. The protein content (mol %) was calculated from emPAI: protein content (mol %) = emPAI/∑ (emPAI) × 100 [55].

### 4.6. Ribosome Purification

Ribosomes were purified as described in [27] with modifications. *Listeria* cells were grown in BHI to OD_600nm_ = 1.0, harvested by centrifugation, frozen in liquid nitrogen and mechanically lysed in ribosomal buffer [50 mM HEPES, pH 7.5, 50 mM KOAc, 12 mM MgOAc, 1 mM DTT, 1× Protease Inhibitor Cocktail (Biotool, Munich, Germany)] using the laboratory Mixer Mill MM 400 (Retsch, Haan, Germany) with 50 mL jars and grinding balls (25 mm diameter), that were chilled in liquid nitrogen. Ultracentrifugation through a 20% sucrose cushion made of ribosomal buffer was done at >100,000× *g* at 4 °C for 4 h. The ribosomal pellet was washed and stored in the ribosomal buffer. Ribosome concentrations were determined as described in [27] by measuring the absorbance at 260 nm and calculation of the ribosome concentration using an extinction coefficient of 43,500 M^−1^ cm^−1^.

### 4.7. Co-Sedimentation Experiments

Co-sedimentation experiments were performed as described in [27]. Purified ribosomes were prepared as described in 4.6. 4 µM of ribosomes were incubated with 1 µM purified SecA2 and SecA1 as described in the figure legends. Ribosomes were pelleted by ultracentrifugation through a 20% sucrose cushion at >100,000× *g* at 4 °C for 2 h. The supernatant and the pellet fractions were analyzed on Coomassie stained SDS-PAGE gels.

### 4.8. Western Blotting

The amounts of SecA1 and SecA2 proteins were analyzed by western blotting using a StrepTactin-HRP conjugate (IBA lifesciences, Göttingen, Germany). Additionally, a SecA2 rabbit antibody was used for SecA2 detection [56] that was kindly provided by Sven Halbedel (Robert Koch Institute, Wernigerode, Germany).

### 4.9. Polysome Profiling Using Sucrose Gradient Centrifugation

Bacterial cell harvest, lysis and polysome profiling were done according to [57] with slight modifications. The conventional harvest procedure was used, and the resulting frozen cell drops were lysed by the laboratory Mixer Mill MM 400 (Retsch, Haan, Germany) with 50 mL jars and grinding balls (25 mm diameter), that were chilled in liquid nitrogen. Disintegration of the bacteria was realized by milling five times at 15 Hz with chilling of the jars in liquid nitrogen after each milling cycle. Lysed bacteria were collected and stored at −80 °C or directly thawed for preparation of ribosome purification via sucrose gradient ultracentrifugation, which was done as described [57]. The sucrose gradient preparation, fractionation of ribosomal components and absorbance measurements were done using the hybrid instrument Gradient Station^TM^ (BioComp Instruments, Fredericton, NB, Canada).

### 4.10. Isolation of Cytosolic RNA

RNA of eukaryotic cells was isolated using miRNeasy Mini Kit (Qiagen, Hilden, Germany). The lysis of transfected BMDM (bone marrow derived macrophages) cells was done by adding QIAzol. Cells were mixed gently and incubated for 5 min at room temperature (RT). Subsequently, cells were homogenized by adding 0.2 volume of chloroform and incubated for 2 min at RT. The samples were centrifuged at 16,000× *g* at 4 °C for 15 min. The aqueous phase, that contains the RNA, was separated and mixed with 1.5 volumes ethanol absolute (Sigma, Steinheim, Germany). Subsequently, the samples were transferred to the columns supplied with the miRNeasy Mini Kit (Qiagen) and treated according to the manufacturer’s instructions. An on-column DNase (Qiagen) digestion was done in between.

Bacterial cells were pelleted by centrifugation at 16,000× *g* for 3 min and washed with SET buffer, containing 50 mM NaCl, 5 mM EDTA, 10% sodium dodecyl sulfate and 30 mM Tris-HCl pH 7.0. Lysis of bacterial cells was carried out by resuspending bacterial cell pellets in 0.1 mL 50 mM Tris-HCl buffer containing 50 mg/mL lysozyme (Sigma), 25 U of mutanolysin (Sigma), 40 U of SUPERase In^TM^ (Ambion, Austin, TX, USA) and 0.2 mg of proteinase K (Ambion) for 45 min at 37 °C with 350 rpm shaking. QIAzol was added to the bacterial lysates and the RNA was isolated by using the miRNeasy Mini Kit (Qiagen). RNA was eluted in RNase-free water and stored at −80 °C until needed. The Qubit RNA BR Assay Kit (Thermo Fisher Scientific, Langenselbold, Germany) was used for RNA quantification and the quality was analyzed by Nano and Small RNA chips for the Agilent 2100 Bioanalyzer (Agilent Technologies, Santa Clara, CA, USA).

### 4.11. Isolation of Sec-RNA and SecA2/SecA1-Associated RNAs

The supernatant of bacteria cells grown at 37 °C to an OD_600nm_ = 1.0 was obtained by centrifugation at 16,000× *g* for 3 min. The supernatant was sterile filtered using 0.22 µm filter (Merck, Darmstadt, Germany). Extraction of sec-RNA (secreted RNA) was done by adding 1:1 ethanol absolute (Sigma) to cell-free supernatant and overnight precipitation at −20 °C. RNA was collected by centrifugation at 4 °C for 45 min at 6000× *g*. The miRNeasy Mini Kit (Qiagen) was used for RNA isolation.

The SecA2/SecA1 protein solution was incubated with phenol/chloroform isoamylalcohol (Roth) for 5 min at RT. The aqueous phase, that contains the SecA2/SecA1-associated RNA, was separated by centrifugation at 16,000× *g* at 4 °C for 15 min and mixed with 3 M sodium acetate pH 4.8–5.2 (10%) and 1 volume cold ethanol absolute (Sigma) over night at −20 °C. SecA2/SecA1-associated RNA was collected by centrifugation, dried, and resuspended in RNase-free water and stored at −80 °C.

### 4.12. cDNA Synthesis and RNA-Sequencing

First, the RNA samples were treated with RNA 5’ Polyphosphatase (Epicentre, Paris, France). Oligonucleotide adapters were ligated to the 5’ and 3’ ends of the RNA samples. First-strand cDNA synthesis was performed using M-MLV reverse transcriptase and the 3′ adapter as a primer. The resulting cDNAs were amplified by PCR using a high-fidelity DNA polymerase. The cDNA was purified using the Agencourt AMPure XP kit (Beckman Coulter Genomics, Krefeld, Germany). The yield and size distribution of the amplified cDNA were assessed with Agilent DNA High Sensitivity Kit (Agilent, Santa Clara, CA, USA). Libraries were then diluted to 4 nM, pooled, denatured, and further diluted to 10 pM. Sequencing was carried out on the MiSeq using v2 chemistry (Illumina, San Diego, CA, USA).

CLC Genomics workbench 8.5.1 (CLC bio, Qiagen, Hilden, Germany) was then utilized to align processed reads to the reference (NC_003210) and count reads. As parameters for the mapping, a mismatch cost of 2, an insertion cost of 3, a deletion cost of 3 along with a threshold for the length fraction of 0.8 and a required similarity fraction of 0.8 were chosen. The counting was carried out in a strand-specific manner. Sequences originating from this study are deposited in “JLUbox”. Link: https://jlubox.uni-giessen.de/filestable/MlJ2NnBvRmZMU0V5Y0d1RUM2V2ZL, accessed on 1 September 2022.

### 4.13. Cell Culture and Transfection Assay

Bone marrow-derived macrophages (BMDMs) were isolated from mouse strain C57BL/6 and cultured for six days in RPMI 1640 medium (Thermo Fisher Scientific) supplemented with 10% fetal calf serum (FCS, Biochrom, Berlin, Germany), 100 U/mL penicillin, 100 µg/mL streptomycin and 30% supernatant of L929 cells prior transfection. BMDM cells (10^6^ cells) were transfected with 100 ng of isolated bacterial RNA, complexed with lipofectamine 2000 according to manufacturer’s instructions (Thermo Fisher Scientific).

### 4.14. Quantitative Real-Time PCR

Eukaryotic RNA (500 ng) was treated with DNase I (Qiagen, Hilden, Germany) followed by reverse transcription to generate complementary DNA (cDNA) via SuperScript II reverse transcriptase (Thermo Fisher Scientific, Langenselbold, Germany). The QuantiTect SYBR Green PCR Kit (Qiagen) was used for cDNA amplification of IFN-β by the QuantiTect primers Mm_Ifnb1_1_SG (Qiagen) using StepOnePlus Real-Time PCR System (Applied Biosystems, Darmstadt, Germany).

### 4.15. Statistical Analysis

Statistical analysis of experiments was performed with SigmaPlot 11 (Systat Software). *p* values of ≤0.05, ≤0.01, and ≤0.001 were considered statistically significant. The number of individual experiments (at least n = 3) is indicated in the Section 2. Data are shown as the mean ± standard deviation (SD).

## Figures and Tables

**Figure 1 ijms-23-15021-f001:**
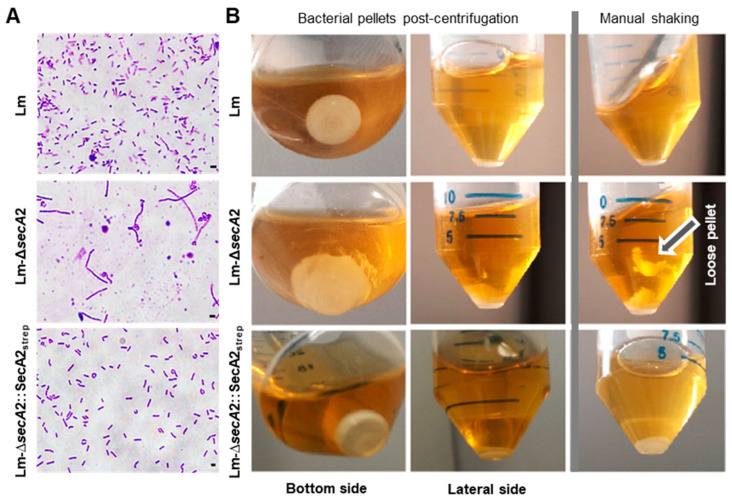
(**A**) Assessment of SecA2 functionality in Lm-∆*secA*2::SecA2_strep_ (*Lm* = *L. monocytogenes*). The characteristic cell chain growth of the deletion mutant Lm-Δ*secA*2 is restored to single cell morphology of the wild type in the complemented strain Lm-∆*secA*2::SecA2_strep_; and (**B**) SecA2 complementation in Lm-∆*secA*2::SecA2_strep_ restored consistence of the pellet. In contrast to Lm-∆*secA2* pellet which is unstable and detaches easily upon being shaken the pellet of Lm-∆*secA*2::SecA2_strep_ resembles that obtained with the wild type strain.

**Figure 2 ijms-23-15021-f002:**
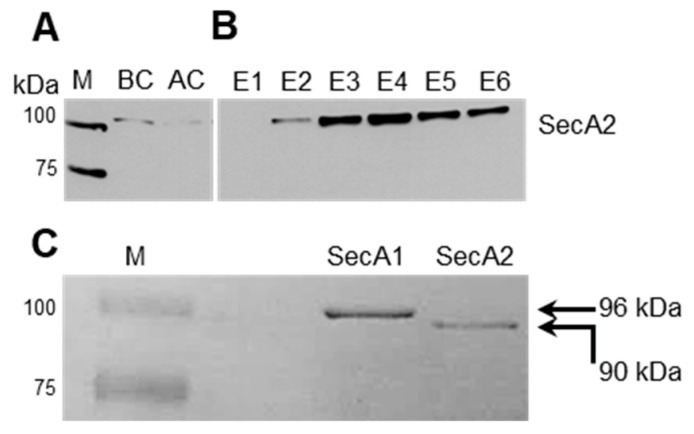
Affinity chromatography for SecA2. (**A**,**B**) Identification of SecA2 in cytosolic fractions BC (before column), AC (after column) and elution fractions (E1–E6) using specific anti-SecA2 antibody; and (**C**) comparison of affinity chromatography purified Strep-tagged SecA1 (96 kDa) and SecA2 (90 kDa). M = Size ladder.

**Figure 3 ijms-23-15021-f003:**
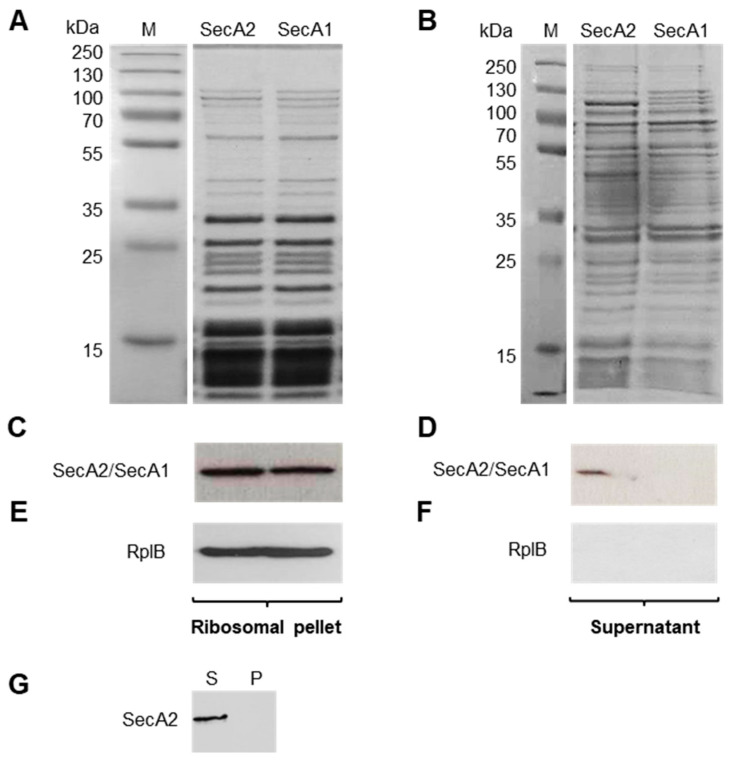
Co-sedimentation experiments of SecA2 with ribosomes. Purified SecA2 and SecA1 proteins (1 µM) were mixed with purified ribosomes (4 µM) and sedimented by sucrose cushion ultracentrifugation. Coomassie stained gel of ribosomal pellet (**A**) and the corresponding supernatant (**B**) are depicted. Western blot analyses were used to detect SecA2 and SecA1 in the ribosomal pellet (**C**) and in the supernatant (**D**); (**E**,**F**) detection of the ribosomal protein RplB in the ribosomal pellet and the supernatant; and (**G**) SecA2 sedimentation in the pellet (P) and supernatant (S) fractions in the absence of ribosomes.

**Figure 4 ijms-23-15021-f004:**
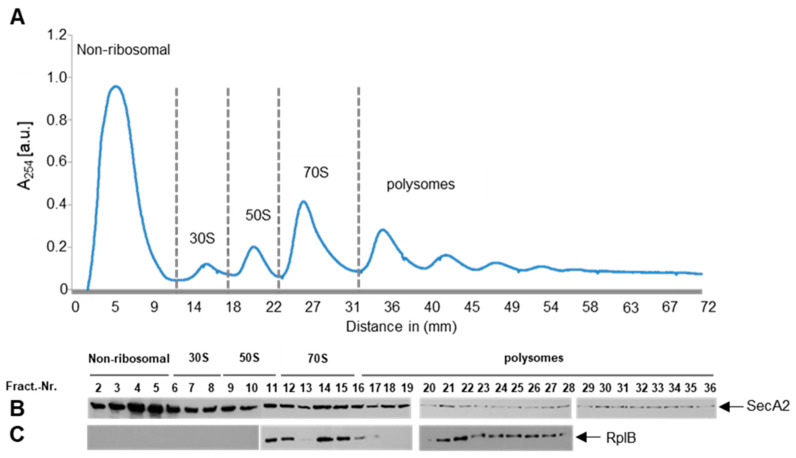
Detection of SecA2 on polysomes. (**A**) the ribosome profile of Lm-∆*secA*2::SecA2_strep_ is depicted. The first gradient fractions show free ribosome-unbound low molecular weight proteins. The ribosomal components 30S, 50S, 70S and polysomes were detected at higher sucrose concentration; and (**B**,**C**) Detection of SecA2 and the ribosomal protein RplB in different gradient fractions. Ribosomal peaks were normalized to the same gradient baseline by subtracting the area under (a.u.) the polysome profile for quantitative comparison.

**Figure 5 ijms-23-15021-f005:**
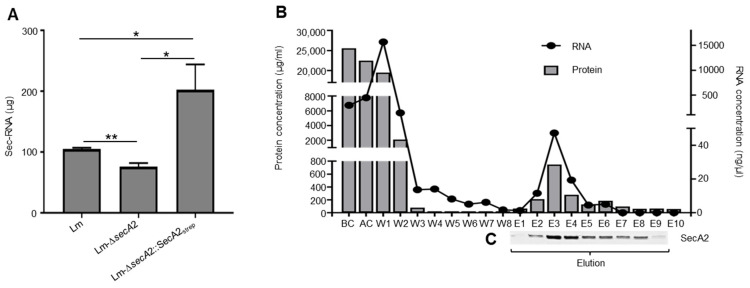
(**A**) Differences in the amounts of secreted RNA (sec-RNA) isolated from the supernatant of wild type Lm, Lm-∆*secA2* and Lm-∆*secA2*::SecA2_strep_; (**B**) correlation between SecA2 protein concentration and its co-eluted RNA. Protein and RNA concentration of cytosolic fractions before (BC), after (AC) passing through the streptactin chromatography column, in wash steps (W1-W8) and elution fractions (E1-E10); and (**C**) the corresponding coomassie gel for SecA2 protein. Data are presented as means ± SD of results from three experiments * *p* ≤ 0.05; ** *p* ≤ 0.01.

**Figure 6 ijms-23-15021-f006:**
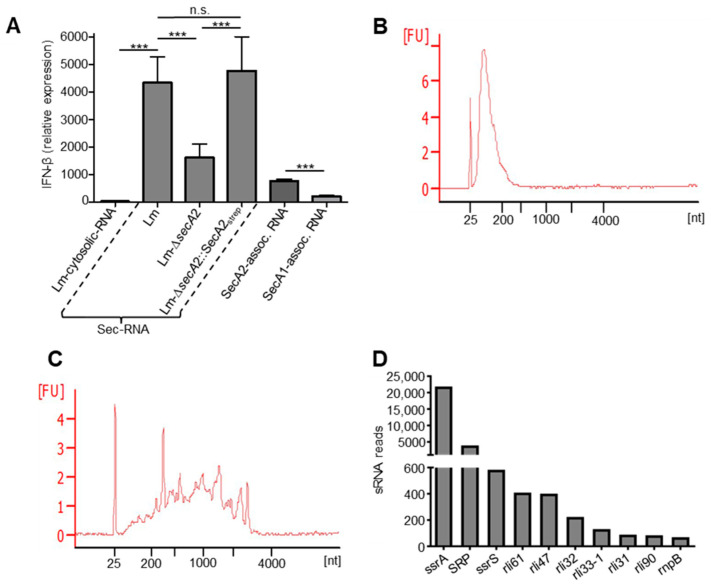
(**A**) IFN-β induction by cytosolic RNA, secreted RNA from wild type cultures (Lm-sec-RNA), SecA2 deletion mutant (Lm-∆*secA2*), the complemented strain (Lm-∆*secA2*::SecA2_strep_) and SecA2- and SecA1-associated RNAs. Transfection of SecA2-associated RNAs (SecA2-assoc. RNA) in bone marrow-derived macrophages (BMDM) showed higher IFN-β induction capability as compared to SecA1-associated RNAs (SecA1-assoc. RNA). The highest IFN-β induction was shown by transfection of Lm-sec-RNA isolated from the supernatant of wild type Lm cultures. Cytosolic RNA (Lm-cytosolic-RNA) was the weakest in terms of IFN-β induction. The data represent the mean ± SD of three independent experiments (n.s., nonsignificant; *** *p* < 0.001); (**B**,**C**) bioanalyzer electropherograms (RNA 6000 Nano Chip) showing profiles of RNAs associated to purified SecA2 (**B**) and SecA1 (**C**). The results represent one of three independent experiments; and (**D**) SecA2-associated small non-coding RNAs (sRNAs) arranged according to their transcript numbers (top ten). Data represent means of two independent experiments.

**Figure 7 ijms-23-15021-f007:**
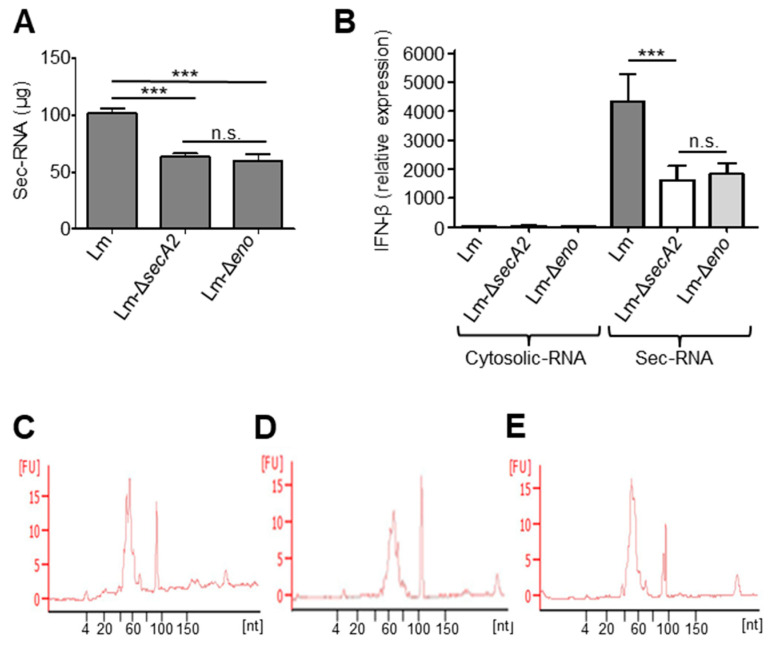
(**A**) Reduced amount of secreted RNA (sec-RNA) in the supernatant of Lm-∆*secA*2 and Lm-∆*eno* as compared to wild type (Lm); (**B**) IFN-β induction capability of cytosolic RNA and sec-RNA isolated from Lm, Lm-∆*secA*2 and Lm-Δ*eno*. No significant differences were detectable for cytosolic RNAs. Lack of enolase and SecA2 reduced significantly the capability of sec-RNA to induce IFN-β response in BMDMs as compared to sec-RNA of Lm; and (**C**–**E**) similar profiles of sec-RNA fractions of Lm (**C**), Lm-∆*secA*2 (**D**) and Lm-Δ*eno* (**E**) analysed by small RNA chip of the Agilent Bioanalyzer. Data are presented as means ± SD of results from three experiments (n.s., nonsignificant; *** *p* < 0.001).

**Figure 8 ijms-23-15021-f008:**
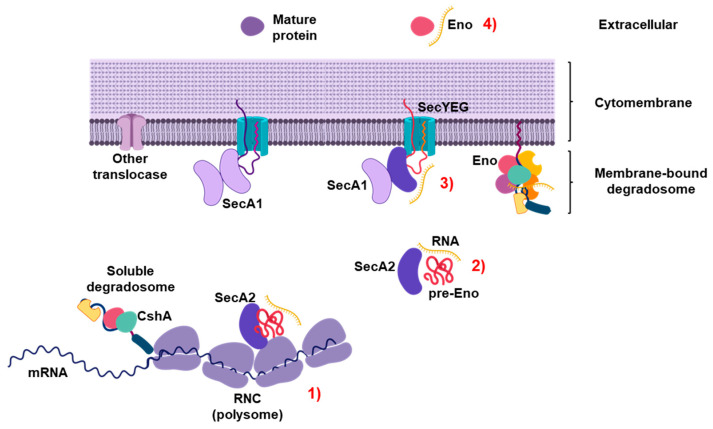
A model of SecA2-dependent secretion of proteins and RNAs in *L. monocytogenes*. (1, 2) SecA2 interacts with translating ribosomes and thereby recognizes SecA2-dependent proteins co-translationally at ribosome-nascent chain complexes (RNC). (3) SecA2 associates with the canonical SecYEG/SecA1 channel. It is suggested that SecA2 either forms homodimers like SecA1 or SecA1 supports SecA2-dependent translocation by heterodimerization with SecA2. (4) The SecA2-dependent protein enolase is a part of the RNA degradosome close to RNCs or at the inner cytomembrane and binds individual RNAs e.g., rli32. SecA2 targets enolase to the SecYEG channel and facilitates the translocation of enolase together with its associated sRNAs in the supernatant.

**Table 1 ijms-23-15021-t001:** Proteins co-isolated with SecA2. All proteins that have been described to be SecA2-dependent [5,16,30] are highlighted in blue. Glycolytic proteins are underlined.

Lmo Nr.	Gene Name	Protein Name	Function (COGs) *	emPAI *	mol % *
lmo0583	secA2	Protein translocase subunit SecA 2	Intracellular trafficking and secretion	5.81	6.75
lmo2632	rplC	50S ribosomal protein L3	Translation	4.75	4.69
lmo1934	hup	DNA-binding protein HU	Replication. recombination and repair	3.54	4.69
lmo2629	rplB	50S ribosomal protein L2	Translation	3.20	3.87
lmo2653 *	tuf	Elongation factor Tu	Translation	2.78	3.69
lmo1830		Short-chain dehydrogenase	Lipid transport and metabolism; Secondary metabolites biosynthesis. transport and catabolism; General function prediction only;	1.73	2.41
lmo1783	rplT	50S ribosomal protein L20	Translation	2.08	2.26
lmo1072	pycA	Pyruvate carboxylase	Energy production and conversion	2.08	2.20
lmo1356		Acetyl-CoA carboxylase biotin carboxyl carrier protein	Lipid transport and metabolism	1.70	2.16
lmo1330	rpsO	30S ribosomal protein S15	Translation	2.32	2.15
lmo2624	rpmC	50S ribosomal protein L29	Translation	1.60	2.09
lmo2016	cspB	Cold shock-like protein CspLB	Transcription	1.51	1.98
lmo1473 *	dnaK	Chaperone protein DnaK	Posttranslational modification. protein turnover. chaperones	1.72	1.96
lmo2631	rplD	50S ribosomal protein L4	Translation	1.53	1.78
lmo2459	gap	Glyceraldehyde-3-phosphate dehydrogenase	Carbohydrate transport and metabolism	1.25	1.60
lmo1596	rpsD	30S ribosomal protein S4	Translation	1.47	1.37
lmo1055	PdhD	Dihydrolipoamide dehydrogenase	Energy production and conversion	0.91	1.33
lmo2625	rplP	50S ribosomal protein L16	Translation	1.00	1.29
lmo1542	rplU	50S ribosomal protein L21	Translation	0.93	1.13
lmo2455 *	eno	Enolase	Carbohydrate transport and metabolism	0.77	1.10
lmo1658	rpsB	30S ribosomal protein S2	Translation	0.94	1.09
lmo0972	dltC	D-alanine--poly(phosphoribitol) ligase subunit 2	Cell wall/membrane biogenesis	0.98	1.02
lmo2613	rplO	50S ribosomal protein L15	Translation	0.75	0.90
lmo2785	kat	Catalase	Inorganic ion transport and metabolism	0.65	0.90
lmo1657	tsf	Elongation factor Ts	Translation	0.70	0.88
lmo2654	fus	Elongation factor G	Translation	0.71	0.78
lmo2630	rplW	50S ribosomal protein L23	Translation	0.62	0.77
lmo2969	groES	10 kDa chaperonin	not in COGs	0.46	0.75
lmo2483	hprK	Phosphocarrier protein HPr	Signal transduction mechanisms	0.70	0.73
lmo1920		Copper chaperone CopZ	Function unknown	0.52	0.64
lmo1364	cspL	Cold shock protein CspA	Transcription	0.47	0.61
lmo0044	rpsF	30S ribosomal protein S6	Translation	0.47	0.61
lmo2608	rpsM	30S ribosomal protein S13	Translation	0.39	0.51
lmo2068 *	groEL	60 kDa chaperonin	Posttranslational modification. protein turnover. chaperones	0.40	0.50
lmo2703		similar to B. subtilis YaaK protein	Function unknown	0.39	0.48
lmo2182		Iron ABC transporter ATP-binding protein	Inorganic ion transport and metabolism; Coenzyme transport and metabolism;	0.37	0.46
lmo1797	rpsP	30S ribosomal protein S16	Translation	0.39	0.44
lmo0866	cshA	DEAD-box ATP-dependent RNA helicase CshA	Replication. recombination and repair; Transcription; Translation;	0.32	0.41
lmo2219	prsA2	Foldase protein PrsA 2	Posttranslational modification. protein turnover. chaperones	0.30	0.34
lmo1398	recA	Protein RecA	Replication. recombination and repair	0.30	0.34
lmo0249	rplA	Ribosomal protein (Fragment)	Translation	0.26	0.34
lmo2511	hpf	Ribosome hibernation promoting factor	Translation	0.26	0.32
lmo1373		2-oxoisovalerate dehydrogenase	Energy production and conversion	0.30	0.31
lmo1388 *	tcsA	CD4+ T-cell-stimulating antigen	General function prediction only	0.25	0.30
lmo0237	gltX	Glutamate--tRNA ligase	Translation	0.20	0.29
lmo1785	infC	Initiation factor IF-3	Translation	0.28	0.29
lmo1267	tig	Trigger factor	Posttranslational modification. protein turnover. chaperones	0.19	0.28
lmo1054 *	pdhC	Pyruvate dehydrogenase E2 component (Dihydrolipoamide acetyltransferase)	Energy production and conversion	0.20	0.26
lmo2615	rpsE	30S ribosomal protein S5	Translation	0.19	0.25
lmo0135		Putative oligopeptide ABC transporter. OppA	Amino acid transport and metabolism	0.21	0.24
lmo2457	tpiA	Triosephosphate isomerase	Carbohydrate transport and metabolism	0.21	0.24
lmo1579		Alanine dehydrogenase	Amino acid transport and metabolism	0.16	0.21
lmo1331	pnpA	Polyribonucleotide nucleotidyltransferase	Translation	0.19	0.21
lmo2799		GTP-binding protein YchF	Carbohydrate transport and metabolism	0.23	0.21
lmo1664	metK	S-adenosylmethionine synthase	Coenzyme transport and metabolism	0.20	0.20
lmo1439 *	sodA	Superoxide dismutase	Inorganic ion transport and metabolism	0.13	0.20
lmo2597	rplM	50S ribosomal protein L13	Translation	0.17	0.20
lmo2510	secA	Protein translocase subunit SecA 1	Intracellular trafficking and secretion	0.21	0.19
lmo2618	rpsH	30S ribosomal protein S8	Translation	0.16	0.19
lmo0539		Tagatose 1.6-diphosphate aldolase	Carbohydrate transport and metabolism	0.13	0.18
lmo2637 *		FMN-binding domain protein	Function unknown	0.13	0.16
lmo2367	pgi	Glucose-6-phosphate isomerase	Carbohydrate transport and metabolism	0.11	0.16
lmo1571	pfkA	ATP-dependent 6-phosphofructokinase	Carbohydrate transport and metabolism	0.12	0.15
lmo0223	cysK	Cysteine synthase	Amino acid transport and metabolism	0.12	0.15
lmo2620	rplE	50S ribosomal protein L5	Translation	0.12	0.15
lmo0416		Uncharacterized protein. DNA-binding. putative regulator	Transcription	0.14	0.14
lmo1052	pdhA	Pyruvate dehydrogenase (Acetyl-transferring) E1 component. alpha subunit	Energy production and conversion	0.10	0.13
lmo2458	pgk	Phosphoglycerate kinase	Carbohydrate transport and metabolism	0.09	0.13
lmo2206	clpB	Chaperone protein ClpB	Posttranslational modification. protein turnover. chaperones	0.12	0.13
lmo1357		acetyl-CoA carboxylase biotin carboxylase subunit	Lipid transport and metabolism	0.12	0.11
lmo1217		Putative aminopeptidase. M42 family	Carbohydrate transport and metabolism	0.09	0.11
lmo1928	aroF	Chorismate synthase	Amino acid transport and metabolism	0.08	0.10
lmo2559	pyrG	CTP synthase	Nucleotide transport and metabolism	0.09	0.09
lmo2456	pgm	2.3-bisphosphoglycerate-independent phosphoglycerate mutase	Carbohydrate transport and metabolism	0.07	0.09
lmo1938	rpsA	30S ribosomal protein S1 homolog	Translation	0.08	0.08
lmo1755	gatA	Glutamyl-tRNA(Gln) amidotransferase subunit A	Translation	0.07	0.08
lmo0727	glmS	Glutamine--fructose-6-phosphate aminotransferase [isomerizing]	Cell wall/membrane biogenesis	0.07	0.08
lmo1286	parE	DNA topoisomerase 4 subunit B	Replication. recombination and repair	0.06	0.07
lmo0258 *	rpoB	RNA polymerase beta subunit	Transcription	0.05	0.06
lmo1003	ptsI	Phosphoenolpyruvate-protein phosphotransferase	Carbohydrate transport and metabolism	0.05	0.06
lmo1634 *	lap	Aldehyde-alcohol dehydrogenase	Energy production and conversion	0.05	0.06
lmo0259 *	rpoC	DNA-directed RNA polymerase subunit beta’	Transcription	0.05	0.05
lmo1504	alaS	Alanine—tRNA ligase	Translation	0.04	0.05
lmo2019	ileS	Isoleucine—tRNA ligase	Translation	0.03	0.04
lmo0007	gyrA	DNA gyrase subunit A	Replication. recombination and repair	0.03	0.03

* COGs = (Clusters of Orthologous Groups of proteins); emPAI = (Exponentially modified protein abundance Index); (mol %) = emPAI/∑ (emPAI) ×100 (57).

## Data Availability

Not applicable.

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
