# Peer review of "SecA2 Associates with Translating Ribosomes and Contributes to the Secretion of Potent IFN-β Inducing RNAs"

_ijms, 2022, doi:10.3390/ijms232315021_

Round 1
Reviewer 1 Report
In SecA2 associates with actively translating ribosomes and contributes to the secretion of potent IFN-b inducing RNAs, Teubner et al. investigate Listeria monocytogenes SecA2. They find it binds to ribosomes and specifically investigate the contribution of SecA2 associated sRNA to IFNb production. Finally, authors investigate enolase as a potential mechanism of SecA2 dependent sRNA secretion and targeting. Collectively, the work presented herein is novel and would advance what is known in the field about secreted RNA and how they impact the host response. However, throughout the manuscript experimental design needs additional explanation, and in some cases, experiments should be reconsidered. At minimum, proteomic and transcriptomic data needs replicates in order to properly justify the claims made by authors in the paper. Additionally, most results sections could be expanded for clarity as outlined below.
Major critiques:
· Figures 2.1 and 2.2 are largely proof of methods and can be moved to supplemental
· 2.5. Previous studies only identify 19 proteins that bind to SecA2. Can authors speculate as to why they are getting 85 proteins and only 11/19 of the previous hits? Did the previous study use cultures grown to an OD600 of 1.0 or at another growth phase?
· An explanation of how the SecA2 co-elution was performed is necessary. Did authors use purified SecA2 and identify every protein bound to purified SecA2 or did they use the strep tagged strain overexpressing SecA2? If the overexpression strain was used, an empty vector control should be performed to identify proteins pulled down by the plasmid. No mention of replicates.
· Authors should consider adding table S2 to the manuscript and adding table 1 to supplement. Table 1 highlights proteins identified in another study where S2 documents the proteins identified in this work. The same highlighting of overlapping proteins can be performed with the larger table.
o In regard to table S2, there is no mention in the manuscript or methods of how COGs were determined. Authors should also define emPAI and mol %.
· Authors speculate in 254-255 that increased RNA may be due to increased SecA2 expression in the complement strain. qPCR validation of SecA2 expression should be done to confirm this overexpression.
· It is unclear how authors are deriving either SecA1 or SecA2 secreted RNA in 2.7. Is it from one eluted fraction? How many replicates were performed? Authors should be comparing the wild-type secreted RNA profile to that of a secA2 mutant and complement strain to determine which RNA are secA2 dependent and their subsequent IFNb response.
· Section 2.8 what samples were used to perform RNA seq? In what replicate? Line 299 says there are a “high proportion” of sRNA in SecA2 associated RNAs. What is that proportion in comparison to? The reported RNAseq data includes expression value but no comparison or fold change to the total secreted RNA from wild-type LM. Also it appears to only be one replicate. These data should be performed in duplicate or triplicate for reproducibility.
· Authors state in 2.8 the 4 highest read numbers of sRNA are ssrA, the 4.5S RNA subunit and ssrS. This is not surprising as these are the most highly transcribed sRNA that Gram positives encode. This result is likely do to a higher abundance of these sRNA than their increased association with SecA2. Again, a comparison of expression to that of the secreted wild type fraction is necessary to make this claim.
· The data on the enolase mutant is strong. Having the same experiments with the secA2 mutant are necessary to attribute the IFNb production and expression of these particular sRNA to secA2 specifically.
Minor critiques:
· Line 46 remove “on the one hand”
· Lines 126-131 state E. coli SecA1 has been shown to bind translating ribosomes. Authors then state similarity between LM SecA1 and SecA2. Has it previously been shown that SecA1 from LM binds translating ribosomes or is this the first report of this phenomenon?
Author Response
Reviewer 1:
Comments and Suggestions for Authors
In SecA2 associates with actively translating ribosomes and contributes to the secretion of potent IFN-b inducing RNAs, Teubner et al. investigate Listeria monocytogenes SecA2. They find it binds to ribosomes and specifically investigate the contribution of SecA2 associated sRNA to IFNb production. Finally, authors investigate enolase as a potential mechanism of SecA2 dependent sRNA secretion and targeting. Collectively, the work presented herein is novel and would advance what is known in the field about secreted RNA and how they impact the host response. However, throughout the manuscript experimental design needs additional explanation, and in some cases, experiments should be reconsidered. At minimum, proteomic and transcriptomic data needs replicates in order to properly justify the claims made by authors in the paper. Additionally, most results sections could be expanded for clarity as outlined below.
Major critiques:
- Comment. Figures 2.1 and 2.2 are largely proof of methods and can be moved to supplemental
Response. We moved figure 2.1 (2A) in supplemental. In our opinion figure 1 is important to demonstrate the functionality of SecA2. Therefore, we prefer to keep it in the manuscript.
- Comment. 2.5. Previous studies only identify 19 proteins that bind to SecA2. Can authors speculate as to why they are getting 85 proteins and only 11/19 of the previous hits? Did the previous study use cultures grown to an OD600 of 1.0 or at another growth phase?
Response. In the previous study of Lenz et al (2003), which we refer to, the bacteria were grown in LB medium (not in BHI as done in our study) till the exponential growth phase (no OD value ist mentioned in the paper). Therefore, different growth conditions can be a reason for the differences in the detected proteins. In addition, the 19 proteins identified by Lenz et al. were detected by separation of supernatant and cell wall proteins on SDS-PAGE. The authors compared the protein patterns of wild type and ∆secA2 deletion mutant on SDS-PAGE. The different protein bands were excised from the gel and identified. In our study we identified proteins that were co-purified with SecA2 (in solution) by affinity chromatography. We think that the combination of different growth media and preparation methods causes the differences.
Reference:
Lenz L. L.; Mohammadi S.; Geissler A.; Portnoy D .A. SecA2-dependent secretion of autolytic enzymes promotes Listeria monocytogenes pathogenesis. Proc Natl Acad Sci U S A 2003. 100:12432–12437. http://doi.org/10.1073/pnas. 2133653100.
- Comment. An explanation of how the SecA2 co-elution was performed is necessary. Did authors use purified SecA2 and identify every protein bound to purified SecA2 or did they use the strep tagged strain overexpressing SecA2? If the overexpression strain was used, an empty vector control should be performed to identify proteins pulled down by the plasmid. No mention of replicates.
Response. We used the complemented strain „Lm-∆secA2::SecA2strep“ that produces Strep-tagged SecA2. SecA2 and its potential binding partners were eluted using affinity chromatography column (4.4. Protein purification). The elution fractions E1 to E6, in which the presence of SecA2 was confirmed via specific antibody (Fig. 2B), were pooled and the proteins were detected using tandem mass spectrometry. We included the information about the use of pooled E1-E6 for the identification of SecA2 co-eluted proteins in part 2.5 of the revised manuscript. No proteins are detectable by affinity chromatography when empty vector control (without SecA2) was used in Lm-∆secA2 deletion mutant. Additionally, neither proteins nor RNAs were eluted from wild type cytosolic lysates, which produces non-tagged SecA2 (Fig. S3).
- Comment. Authors should consider adding table S2 to the manuscript and adding table 1 to supplement. Table 1 highlights proteins identified in another study where S2 documents the proteins identified in this work. The same highlighting of overlapping proteins can be performed with the larger table.
Response. Tabel S2 is added in the revised manuscript (referred to as table 1). As suggested by the reviewer, table 1 was moved in supplemental (Table S1).
Comment. In regard to table S2, there is no mention in the manuscript or methods of how COGs were determined. Authors should also define emPAI and mol %.
Response. The missed explanation are included in the revised manuscript.
- Comment. Authors speculate in 254-255 that increased RNA may be due to increased SecA2 expression in the complement strain. qPCR validation of SecA2 expression should be done to confirm this overexpression.
Response. The expression of SecA2 in the wild-type and the complemented strain were determined using qPCR. The results are included in the revised manuscript (Fig. S3).
- Comment. It is unclear how authors are deriving either SecA1 or SecA2 secreted RNA in 2.7. Is it from one eluted fraction? How many replicates were performed? Authors should be comparing the wild-type secreted RNA profile to that of a secA2 mutant and complement strain to determine which RNA are secA2 dependent and their subsequent IFNb response.
Response. The sec-RNA was isolated from the supernatants of bacterial cultures grown at 37°C to an OD600nm =1.0 (the procedure is described in 4.11). All experiments represent data of at least three independent experiments with the exception of RNA seq data (Fig. 6D), which represent two replicates. As suggested by the reviewer, we complemented fig. 6A. It shows a comparison of wild-type secreted RNA to that of the secA2 deletion mutant and the complemented strain regarding the ability to induce IFNb response in macrophages. The profiles of sec-RNA of Wt and SecA2 mutant are shown in fig. 7 of the revised manuscript. The information about the replicates is now included in the figure legends.
- Comment. Section 2.8 what samples were used to perform RNA seq? In what replicate? Line 299 says there are a “high proportion” of sRNA in SecA2 associated RNAs. What is that proportion in comparison to? The reported RNAseq data includes expression value but no comparison or fold change to the total secreted RNA from wild-type LM. Also it appears to only be one replicate. These data should be performed in duplicate or triplicate for reproducibility.
Response. The samples used to perform RNA seq were RNAs co-eluted with SecA2 by affinity chromatography (two replicates). The primary aim of the RNA Seq was to identify the SecA2 co-eluted RNAs. Indeed the reported RNA seq data describe only the proportion (amount of transcripts) of the SecA2 co-eluted RNA and include no comparison or fold change. A comparison or fold change of SecA2 co-eluted RNA to the total secreted RNA from wild-type Listeria monocytogenes is, in our opinion, misleading due to the completely different RNA isolation procedures. For the isolation of secreted RNA, we grow the bacteria and isolate the secreted RNA from the supernatant. In the case of SecA2 co-eluted RNA, we use affinity chromatography to isolate SecA2 from the lysed cytosolic fraction, then we extract the RNA from SecA2 elution fractions using phenol/chloroform/isoamylalcohol. In our previously published study (Frantz et al. 2019), in which we identified the total secreted RNA from L. monocytogenes, a fold change was calculated between secreted RNA and cytosolic RNA as both were isolated from the same Listeria cultures. Although a comparison is possible, and we have the data of total secreted RNA from wild-type, but as mentioned above, a comparison or fold change will be not helpful for any interpretation.
We mofdified title of section 2.8 „Enrichment of small non-coding RNAs in SecA2-associated RNAs“. In the revised manuscript we use „Detection“ instead of „Enrichment“.
- Comment. Authors state in 2.8 the 4 highest read numbers of sRNA are ssrA, the 4.5S RNA subunit and ssrS. This is not surprising as these are the most highly transcribed sRNA that Gram positives encode. This result is likely do to a higher abundance of these sRNA than their increased association with SecA2. Again, a comparison of expression to that of the secreted wild type fraction is necessary to make this claim.
Response. We agree with the reviewer that the first three sRNAs (ssrA, 4.5S and ssrS) belong to the most abundant sRNAs. We explained in the prior response (see above) why we did not compare the expression of SecA2 co-eluted RNA to the secreted RNA fraction.
- Comment. The data on the enolase mutant is strong. Having the same experiments with the secA2 mutant are necessary to attribute the IFNb production and expression of these particular sRNA to secA2 specifically.
Response. The data on enolase include now the experiments with secA2 mutant (Fig. 7A and 7B)
Minor critiques:
- Comment. Line 46 remove “on the one hand”
Response. Amended as suggested
- Comment. Lines 126-131 state E. coli SecA1 has been shown to bind translating ribosomes. Authors then state similarity between LM SecA1 and SecA2. Has it previously been shown that SecA1 from LM binds translating ribosomes or is this the first report of this phenomenon?
Response. To the best of our knowledge, the binding of SecA1to translating ribosomes in listeria has not been reported yet.
Reviewer 2 Report
Mobarak Mraheil and Trinad Chakraborty group previously described that Listeria monocytogenes (Lm) can secrete small regulatory RNAs (sRNAs) upon infection of eukaryotic cells, which trigger the induction of interferon beta (IFN-β) (Refs. 21 and 22 in the MS). The group now aims to build upon their prior work by addressing the specific participation of SecA2 ATPase in sRNAs secretion. Considering that secA2 is present only in a subset of gram-positive bacteria, SecA2 is involved in protein secretion, and that it is a non-essential gene, the working hypothesis is plausible and worth testing it. Next to its interest for microbial pathogenesis, describing a novel mechanism for sRNA secretion represents an exciting topic for microbiology in general. Therefore, the overall idea of the paper is interesting and justified.
To demonstrate the participation of SecA2 in sRNA secretion in Lm Teubner et al fused a Strep tag to SecA2, overexpressed it in a ΔsecA2 background (Help promoter), and identified the sRNAs that co-eluted with Strep-SecA2 after affinity purification. Among these, they found Rli32, which they showed in their prior work to be a secreted sRNA that strongly induced IFN-β in mammalian cells. Next to it, the authors performed a series of biochemical analyses to show the interaction of SecA2 with ribosomes (in vitro co-sedimentation of SecA2 with ribosomes, ribosome profiling, identification of proteins co-eluting with SecA2 after affinity purification). Finally, the authors showed that some sRNAs secreted by a deletion mutant in eno (enolase, a moonlighting RNA-binding protein that co-eluted with strep-SecA2) were different to those of the wild type.
In my opinion, this work could not convincingly demonstrate the two conclusions stated in the title. Although the experimental approaches used by the authors are adequate the design lacks some specific appropriate negative controls necessary to reveal the background signal which would have been helpful to interpret the results.
1. Co-sedimentation experiments (Fig. 3). SecA2 shows a partial co-sedimentation with ribosomes. To show that there is no cross-contamination with the supernatant it would have been good to have also used a different purified protein that does not bind ribosomes (e.g., a strep-tagged non-bacterial protein like GFP).
2. Ribosome profiling (Fig. 4). Western blot revealed a SecA2 band in all fractions, being the most intense in the ribosome-free one. Considering that strep-tagged SecA2 expression is driven by super-strong P-Help promoter, it would have been good to see the partition in the sucrose gradient of an unrelated protein over-expressed at the same level.
3. Quantification of secreted RNA co-eluted with strep-tagged SecA2 (Fig. 5). To show that SecA2 is specifically involved in sRNA secretion, one would have expected to also see the quantification of the RNA co-eluted with strep-tagged SecA1.
4. Induction of IFN-β by SecA2-associated RNAs (Fig. 6A). Secreted RNA fraction of Lm induced a potent IFN-β induction. If SecA2 is involved in the secretion of IFN-β-inducing sRNAs it would have been good to assess IFN-β induction by secreted sRNA fraction of ΔsecA2 and ΔsecA2::secA2Strep strains.
5. Identification of RNAs co-eluted with Strep-SecA2 (Fig. 6, Table S3). To claim an enrichment it is necessary to use a control sample, and express the results as a ratio (e.g. log2-fold change), just as when you calculate differentially-expressed genes in comparative transcriptomics. I would suggest using the RNA co-eluted with strep-SecA1, with and unrelated protein (strep-GFP) or even just the RNA from the input fraction (which, at least, would help to discard highly abundant carryover contaminating RNAs).
Minor points.
1. Introduction does not state the scientific question and the objective of the work anywhere. The abstract does it very well though.
2. Table S2. It is not indicated anywhere what emPAI and mol% means.
3. Materials and Methods. The composition of the buffers used in the purification of SecA1 and SecA2 is not indicated. The incubation conditions and the buffer used in co-sedimentation experiments (section 4.7 and legend to Fig. 4) are not indicated. This information is critical to grasp how stringent were the conditions in binding experiments. In the absence of appropriate negative controls (see above) knowing the composition of binding, washing and elution buffers is even more important to interpret the specificity of the identified protein and RNA binders.
4. Table S3 and Fig 6D. To the best of my knowledge, the number of reads depends on the length of the gene. It would be better to use RPKM instead, which is a normalized value. Or even better, as I suggested above, to conduct a comparative quantitative analysis versus a control sample.
5. Section 2.9 (Fig. 7). From what I read in the text, it is not fully clear to me how this set of experiments connects with the rest of the ms.
Author Response
Reviewer 2:
Comments and Suggestions for Authors
Mobarak Mraheil and Trinad Chakraborty group previously described that Listeria monocytogenes (Lm) can secrete small regulatory RNAs (sRNAs) upon infection of eukaryotic cells, which trigger the induction of interferon beta (IFN-β) (Refs. 21 and 22 in the MS). The group now aims to build upon their prior work by addressing the specific participation of SecA2 ATPase in sRNAs secretion. Considering that secA2 is present only in a subset of gram-positive bacteria, SecA2 is involved in protein secretion, and that it is a non-essential gene, the working hypothesis is plausible and worth testing it. Next to its interest for microbial pathogenesis, describing a novel mechanism for sRNA secretion represents an exciting topic for microbiology in general. Therefore, the overall idea of the paper is interesting and justified.
To demonstrate the participation of SecA2 in sRNA secretion in Lm Teubner et al fused a Strep tag to SecA2, overexpressed it in a ΔsecA2 background (Help promoter), and identified the sRNAs that co-eluted with Strep-SecA2 after affinity purification. Among these, they found Rli32, which they showed in their prior work to be a secreted sRNA that strongly induced IFN-β in mammalian cells. Next to it, the authors performed a series of biochemical analyses to show the interaction of SecA2 with ribosomes (in vitro co-sedimentation of SecA2 with ribosomes, ribosome profiling, identification of proteins co-eluting with SecA2 after affinity purification). Finally, the authors showed that some sRNAs secreted by a deletion mutant in eno (enolase, a moonlighting RNA-binding protein that co-eluted with strep-SecA2) were different to those of the wild type.
In my opinion, this work could not convincingly demonstrate the two conclusions stated in the title. Although the experimental approaches used by the authors are adequate the design lacks some specific appropriate negative controls necessary to reveal the background signal which would have been helpful to interpret the results.
- Comment. Co-sedimentation experiments (Fig. 3). SecA2 shows a partial co-sedimentation with ribosomes. To show that there is no cross-contamination with the supernatant it would have been good to have also used a different purified protein that does not bind ribosomes (e.g., a strep-tagged non-bacterial protein like GFP).
Response: The exclusive detection of the ribosomal protein RplB in the ribosomal pellet, but not the supernatant (Fig. 3E, 3F) and the fact that the presence of ribosomes is crucial for the co-sedimentation of SecA2 (Fig. 3G) demonstrate that there is no cross-contamination with the supernatant. We also used SecA1 as a control, because it has been previously shown to bind translating ribosomes.
Reference: Huber D.; Rajagopalan N.; Preissler S.; Rocco M. A.; Merz F.; Kramer G.; Bukau B. SecA interacts with ribosomes in order to facilitate posttranslational translocation in bacteria. Mol Cell 2011. 41:343–353. http://doi.org/10.1016/j.molcel.2010.12.028.
- Comment. Ribosome profiling (Fig. 4). Western blot revealed a SecA2 band in all fractions, being the most intense in the ribosome-free one. Considering that strep-tagged SecA2 expression is driven by super-strong P-Help promoter, it would have been good to see the partition in the sucrose gradient of an unrelated protein over-expressed at the same level.
Response. RplB, which we use as a control, is a highly expressed ribosomal protein. RplB is not detectable in the first six ribosome-free fractions (Fig. 4C).
- Comment. Quantification of secreted RNA co-eluted with strep-tagged SecA2 (Fig. 5). To show that SecA2 is specifically involved in sRNA secretion, one would have expected to also see the quantification of the RNA co-eluted with strep-tagged SecA1.
Response. To show that SecA2 is specifically involved in RNA secretion, we compare the amounts of secreted RNA (sec-RNA) of wild type, Lm-∆secA2 and the complemented ∆secA2 strain. The data show that the deletion of SecA2 reduces the sec-RNA amounts (although Lm-∆secA2 produces SecA1), whereas its overproduction increases the sec-RNA amounts as compared to the wild type (Fig. 5A). This data show that SecA2 is specifically involved in RNA secretion.
SecA1 is an essential protein and therefore not deletable (we have no deletion mutant) and the strep-tagged SecA1is produced in the wild type.
- Comment. Induction of IFN-β by SecA2-associated RNAs (Fig. 6A). Secreted RNA fraction of Lm induced a potent IFN-β induction. If SecA2 is involved in the secretion of IFN-β-inducing sRNAs it would have been good to assess IFN-β induction by secreted sRNA fraction of ΔsecA2 and ΔsecA2::secA2Strep strains.
Response. As suggested by the reviewer, we complemented fig. 6A in the revised manuscript. The IFN-β induction by secreted sRNA fraction of ΔsecA2 and ΔsecA2::secA2Strep strains is included in fig. 6A.
- Identification of RNAs co-eluted with Strep-SecA2 (Fig. 6, Table S3). To claim an enrichment it is necessary to use a control sample, and express the results as a ratio (e.g. log2-fold change), just as when you calculate differentially-expressed genes in comparative transcriptomics. I would suggest using the RNA co-eluted with strep-SecA1, with and unrelated protein (strep-GFP) or even just the RNA from the input fraction (which, at least, would help to discard highly abundant carryover contaminating RNAs).
Response. The primary aim of the RNA Seq was to identify the SecA2 co-eluted RNAs. We agree the reported RNA seq data describe only the proportion (amount of transcripts) of the SecA2 co-eluted RNA and include no comparison or fold change. Therefore, We mofdified title of section 2.8 „Enrichment of small non-coding RNAs in SecA2-associated RNAs“. In the revised manuscript we use „Detection“ instead of „Enrichment“.
A comparison or fold change of SecA2 co-eluted RNA to the total secreted RNA from wild-type Listeria monocytogenes is, in our opinion, misleading due to the completely different RNA isolation procedures. For the isolation of total secreted RNA, we grow the bacteria and isolate the secreted RNA from the supernatant. In the case of SecA2 co-eluted RNA, we use affinity chromatography to isolate SecA2 from the lysed cytosolic fraction, then we extract the RNA from SecA2 elution fractions. In our previously published study (Frantz et al. 2019), in which we identified the total secreted RNA from L. monocytogenes, a fold change was calculated between secreted RNA and cytosolic RNA as both were isolated from the same Listeria cultures. Although a comparison is possible, and we have the data of total secreted RNA from wild-type, but as mentioned above, a comparison or fold change will be not helpful for any interpretation.
Minor points.
Comment. 1. Introduction does not state the scientific question and the objective of the work anywhere. The abstract does it very well though.
We respectfully disagree with the reviewer that the introduction does not state the scientific question and the objective of the work anywhere. Especially the statement “anywhere” confuses us, as the introduction includes all aspects mentioned in the abstract, which is all right in the opinion of the reviewer.
The introduction:
- gives background about the importance of protein secretion and the so far described secretion systems in Listeria monocytogenes
- highlights the importance of the Sec system as the major pathway by which listerial proteins are secreted
- describes the SecA2 pathway
- describes differences between SecA1 and SecA2
- gives information that lack of SecA2 decreases the levels of secreted RNAs in the supernatant and reduces the type I interferon response.
Considering that SecA2 is the main figure in our manuscript, we believe that the introduction covers it in a proper way.
Comment. 2. Table S2. It is not indicated anywhere what emPAI and mol% means.
Response. The revised manuscript includes explanation of what emPAI and mol% mean.
Comment. 3. Materials and Methods. The composition of the buffers used in the purification of SecA1 and SecA2 is not indicated. The incubation conditions and the buffer used in co-sedimentation experiments (section 4.7 and legend to Fig. 4) are not indicated. This information is critical to grasp how stringent were the conditions in binding experiments. In the absence of appropriate negative controls (see above) knowing the composition of binding, washing and elution buffers is even more important to interpret the specificity of the identified protein and RNA binders.
Response. The composition of the buffers used in the purification in included in section 4.4 (Protein purification).
Comment. 4. Table S3 and Fig 6D. To the best of my knowledge, the number of reads depends on the length of the gene. It would be better to use RPKM instead, which is a normalized value. Or even better, as I suggested above, to conduct a comparative quantitative analysis versus a control sample.
Response. We agree with the reviewer. Table S3 includes RPKM values.
Comment. 5. Section 2.9 (Fig. 7). From what I read in the text, it is not fully clear to me how this set of experiments connects with the rest of the ms.
Response. In the manuscript, we describe the contribution of SecA2 in the secretion of RNAs. We found enolase to be one of the SecA2 co-isolated proteins in our MS data. Additionally, enolase is known to be secreted in a SecA2-dependent manner and has the propensity to bind RNA. Therefore, we generated a Lm-Δeno deletion mutant to explore the role of enolase in RNA secretion.
The revised manuscript includes a short introduction to explain the connection of Section 2.9 (enolase) to the rest of manuscript.
Round 2
Reviewer 1 Report
Authors took careful note to address my concerns. I believe the paper is much improved after initial review and advise publication.
My minor comment is in regard to comment 2.5: the authors provide reasoning for the potential discrepancy in their reply to me but speculation should be added to the text as well for the reader, so they don't have to go back to the other papers methods to understand potential differences.
- Comment. 2.5. Previous studies only identify 19 proteins that bind to SecA2. Can authors speculate as to why they are getting 85 proteins and only 11/19 of the previous hits? Did the previous study use cultures grown to an OD600 of 1.0 or at another growth phase?
Response. In the previous study of Lenz et al (2003), which we refer to, the bacteria were grown in LB medium (not in BHI as done in our study) till the exponential growth phase (no OD value ist mentioned in the paper). Therefore, different growth conditions can be a reason for the differences in the detected proteins. In addition, the 19 proteins identified by Lenz et al. were detected by separation of supernatant and cell wall proteins on SDS-PAGE. The authors compared the protein patterns of wild type and ∆secA2 deletion mutant on SDS-PAGE. The different protein bands were excised from the gel and identified. In our study we identified proteins that were co-purified with SecA2 (in solution) by affinity chromatography. We think that the combination of different growth media and preparation methods causes the differences.
Author Response
We thank the reviewer for the constructive comments.
The amended version of the manuscript takes into consideration the recommendationsmade by the reviewer.
- The revised manuscript includes our speculation about the reasons for the differences between SecA2 co-eluted proteins identified in our study and those published previously.
Reviewer 2 Report
Reviewer 2:
Comments and Suggestions for Authors
Mobarak Mraheil and Trinad Chakraborty group previously described that Listeria monocytogenes (Lm) can secrete small regulatory RNAs (sRNAs) upon infection of eukaryotic cells, which trigger the induction of interferon beta (IFN-β) (Refs. 21 and 22 in the MS). The group now aims to build upon their prior work by addressing the specific participation of SecA2 ATPase in sRNAs secretion. Considering that secA2 is present only in a subset of gram-positive bacteria, SecA2 is involved in protein secretion, and that it is a non-essential gene, the working hypothesis is plausible and worth testing it. Next to its interest for microbial pathogenesis, describing a novel mechanism for sRNA secretion represents an exciting topic for microbiology in general. Therefore, the overall idea of the paper is interesting and justified.
To demonstrate the participation of SecA2 in sRNA secretion in Lm Teubner et al fused a Strep tag to SecA2, overexpressed it in a ΔsecA2 background (Help promoter), and identified the sRNAs that co-eluted with Strep-SecA2 after affinity purification. Among these, they found Rli32, which they showed in their prior work to be a secreted sRNA that strongly induced IFN-β in mammalian cells. Next to it, the authors performed a series of biochemical analyses to show the interaction of SecA2 with ribosomes (in vitro co-sedimentation of SecA2 with ribosomes, ribosome profiling, identification of proteins co-eluting with SecA2 after affinity purification). Finally, the authors showed that some sRNAs secreted by a deletion mutant in eno (enolase, a moonlighting RNA-binding protein that co-eluted with strep-SecA2) were different to those of the wild type.
In my opinion, this work could not convincingly demonstrate the two conclusions stated in the title. Although the experimental approaches used by the authors are adequate the design lacks some specific appropriate negative controls necessary to reveal the background signal which would have been helpful to interpret the results.
- Comment. Co-sedimentation experiments (Fig. 3). SecA2 shows a partial co-sedimentation with ribosomes. To show that there is no cross-contamination with the supernatant it would have been good to have also used a different purified protein that does not bind ribosomes (e.g., a strep-tagged non-bacterial protein like GFP).
Response: The exclusive detection of the ribosomal protein RplB in the ribosomal pellet, but not the supernatant (Fig. 3E, 3F) and the fact that the presence of ribosomes is crucial for the co-sedimentation of SecA2 (Fig. 3G) demonstrate that there is no cross-contamination with the supernatant. We also used SecA1 as a control, because it has been previously shown to bind translating ribosomes.
Reference: Huber D.; Rajagopalan N.; Preissler S.; Rocco M. A.; Merz F.; Kramer G.; Bukau B. SecA interacts with ribosomes in order to facilitate posttranslational translocation in bacteria. Mol Cell 2011. 41:343–353. http://doi.org/10.1016/j.molcel.2010.12.028.
>>>> The presence of RplB in the pellet demonstrates that ribosomes are sedimented in these experimental conditions; the exclusive detection of SecA1 in the sediment confirms that SecA1 efficiently binds ribosomes (positive control). It is the negative control what is missing, i.e. a protein that does not bind ribosomes, which should be detected exclusively in the supernatant.
- Comment. Ribosome profiling (Fig. 4). Western blot revealed a SecA2 band in all fractions, being the most intense in the ribosome-free one. Considering that strep-tagged SecA2 expression is driven by super-strong P-Help promoter, it would have been good to see the partition in the sucrose gradient of an unrelated protein over-expressed at the same level.
Response. RplB, which we use as a control, is a highly expressed ribosomal protein. RplB is not detectable in the first six ribosome-free fractions (Fig. 4C).
>>>> RplB is an endogenous abundant protein, which is used here as a marker of large ribosomal subunit partition among fractions. What is missing here is a negative control, that is, an over-expressed protein that does not bind ribosomes whose detection is restricted exclusively to the to fractions 2 to 5.
- Comment. Quantification of secreted RNA co-eluted with strep-tagged SecA2 (Fig. 5). To show that SecA2 is specifically involved in sRNA secretion, one would have expected to also see the quantification of the RNA co-eluted with strep-tagged SecA1.
Response. To show that SecA2 is specifically involved in RNA secretion, we compare the amounts of secreted RNA (sec-RNA) of wild type, Lm-∆secA2 and the complemented ∆secA2 strain. The data show that the deletion of SecA2 reduces the sec-RNA amounts (although Lm-∆secA2 produces SecA1), whereas its overproduction increases the sec-RNA amounts as compared to the wild type (Fig. 5A). This data show that SecA2 is specifically involved in RNA secretion.
SecA1 is an essential protein and therefore not deletable (we have no deletion mutant) and the strep-tagged SecA1is produced in the wild type.
>>>> Where it is written “specifically involved” it is meant that SecA2 belongs to a molecular system specifically involved in RNA secretion. Thus, when a related protein involved in a molecular system that secretes exclusively proteins (ie, SecA1) is overexpressed, the amount of secreted RNA should not change.
>>>> The text does not emphasize that SecA2 is the only system. Thus, this comment might be ignored.
- Comment. Induction of IFN-β by SecA2-associated RNAs (Fig. 6A). Secreted RNA fraction of Lm induced a potent IFN-β induction. If SecA2 is involved in the secretion of IFN-β-inducing sRNAs it would have been good to assess IFN-β induction by secreted sRNA fraction of ΔsecA2 and ΔsecA2::secA2Strep strains.
Response. As suggested by the reviewer, we complemented fig. 6A in the revised manuscript. The IFN-β induction by secreted sRNA fraction of ΔsecA2 and ΔsecA2::secA2Strep strains is included in fig. 6A.
>>>> Fine with the addition.
- Identification of RNAs co-eluted with Strep-SecA2 (Fig. 6, Table S3). To claim an enrichment it is necessary to use a control sample, and express the results as a ratio (e.g. log2-fold change), just as when you calculate differentially-expressed genes in comparative transcriptomics. I would suggest using the RNA co-eluted with strep-SecA1, with and unrelated protein (strep-GFP) or even just the RNA from the input fraction (which, at least, would help to discard highly abundant carryover contaminating RNAs).
Response. The primary aim of the RNA Seq was to identify the SecA2 co-eluted RNAs. We agree the reported RNA seq data describe only the proportion (amount of transcripts) of the SecA2 co-eluted RNA and include no comparison or fold change. Therefore, We mofdified title of section 2.8 „Enrichment of small non-coding RNAs in SecA2-associated RNAs“. In the revised manuscript we use „Detection“ instead of „Enrichment“.
>>>> Fine with the amendment.
A comparison or fold change of SecA2 co-eluted RNA to the total secreted RNA from wild-type Listeria monocytogenes is, in our opinion, misleading due to the completely different RNA isolation procedures. For the isolation of total secreted RNA, we grow the bacteria and isolate the secreted RNA from the supernatant. In the case of SecA2 co-eluted RNA, we use affinity chromatography to isolate SecA2 from the lysed cytosolic fraction, then we extract the RNA from SecA2 elution fractions. In our previously published study (Frantz et al. 2019), in which we identified the total secreted RNA from L. monocytogenes, a fold change was calculated between secreted RNA and cytosolic RNA as both were isolated from the same Listeria cultures. Although a comparison is possible, and we have the data of total secreted RNA from wild-type, but as mentioned above, a comparison or fold change will be not helpful for any interpretation.
>>>> In my comment I have not suggested using “the total secreted RNA from wild-type Listeria monocytogenes” as a control, but “the RNA co-eluted with strep-SecA1, with an unrelated protein (strep-GFP) or even just the RNA from the input fraction”. Proteins can bind RNA to some extent, either within the cell or even after the cell is lyzed, and the detection rate depends on the sensitivity of the technique. The question here is to ascertain which of the RNA molecules identified in the pull down actually bind SecA2 in physiological conditions and which ones are false positives (either because they are very abundant or simply sticky). These results thus revealed SecA2-binding sRNA candidates, which might deserve a further validation.
Minor points.
Comment. 1. Introduction does not state the scientific question and the objective of the work anywhere. The abstract does it very well though.
We respectfully disagree with the reviewer that the introduction does not state the scientific question and the objective of the work anywhere. Especially the statement “anywhere” confuses us, as the introduction includes all aspects mentioned in the abstract, which is all right in the opinion of the reviewer.
The introduction:
- gives background about the importance of protein secretion and the so far described secretion systems in Listeria monocytogenes
- highlights the importance of the Sec system as the major pathway by which listerial proteins are secreted
- describes the SecA2 pathway
- describes differences between SecA1 and SecA2
- gives information that lack of SecA2 decreases the levels of secreted RNAs in the supernatant and reduces the type I interferon response.
Considering that SecA2 is the main figure in our manuscript, we believe that the introduction covers it in a proper way.
>>>>>> In the abstract it is written: “Sec-dependent translocation pathway is a major route for protein secretion in L. monocytogenes, but mechanistic insights into the secretion of RNA by these pathways are lacking.”, which is fine for an abstract. Thus, as I understood from the abstract, the scientific question is “what is the molecular machinery involved in RNA secretion?” and the hypothesis would be that “SecA2 participates in RNA secretion”. The introduction does not provide information to make the reader understand the relevance of this scientific question. For instance, why is it important RNA secretion? why is it important in pathogenic bacteria? is it a molecular mechanism found in other bacteria or just on Listeria monocytogenes? Why would be important to describe the molecular mechanism involved in RNA secretion? The authors jumped directly to justify their hypothesis, but they did not re-visit the scientific question.
Comment. 2. Table S2. It is not indicated anywhere what emPAI and mol% means.
Response. The revised manuscript includes explanation of what emPAI and mol% mean.
>>>>>> Fine with the addition.
Comment. 3. Materials and Methods. The composition of the buffers used in the purification of SecA1 and SecA2 is not indicated. The incubation conditions and the buffer used in co-sedimentation experiments (section 4.7 and legend to Fig. 4) are not indicated. This information is critical to grasp how stringent were the conditions in binding experiments. In the absence of appropriate negative controls (see above) knowing the composition of binding, washing and elution buffers is even more important to interpret the specificity of the identified protein and RNA binders.
Response. The composition of the buffers used in the purification in included in section 4.4 (Protein purification).
>>>>> Fine with the addition.
Comment. 4. Table S3 and Fig 6D. To the best of my knowledge, the number of reads depends on the length of the gene. It would be better to use RPKM instead, which is a normalized value. Or even better, as I suggested above, to conduct a comparative quantitative analysis versus a control sample.
Response. We agree with the reviewer. Table S3 includes RPKM values.
>>>>> Fine with the amendment.
Comment. 5. Section 2.9 (Fig. 7). From what I read in the text, it is not fully clear to me how this set of experiments connects with the rest of the ms.
Response. In the manuscript, we describe the contribution of SecA2 in the secretion of RNAs. We found enolase to be one of the SecA2 co-isolated proteins in our MS data. Additionally, enolase is known to be secreted in a SecA2-dependent manner and has the propensity to bind RNA. Therefore, we generated a Lm-Δeno deletion mutant to explore the role of enolase in RNA secretion.
The revised manuscript includes a short introduction to explain the connection of Section 2.9 (enolase) to the rest of manuscript.
>>>>>> Fine with the addition.
***************************************************************
Overall feedback:
In my opinion, the most relevant findings shown in the current work are in Fig. 6:
· Secreted RNA fraction from the wild type, a secA2 deletion mutant, and the complemented strain induce different levels of IFNbeta induction when the same amount of each of these where transfected in bone marrow-derived macrophages.
· SecA2-associated RNA shows a radically different size pattern and triggers a higher IFNbeta induction compared to SecA1-asssociate ones.
These results indicated that SecA2 binds a specific set of RNAs, and that it is relevant for IFNbeta-inducing sRNA secretion. It is not clear whether it participates directly on sRNA secretion or indirectly, for instance by assembling a ribonucleoparticle relevant for sRNA stability. The decreased IFNbeta response of secRNA from secA2 mutant is consistently reproduced by an enolase mutant, a SecA2 binder and an RNA-binding protein, which underpins the (direct or indirect) participation of SecA2 in IFNbeta-inducing sRNAs secretion.
The exact nature of the IFNbeta-inducing sRNAs whose secretion depends on SecA2 (Section 2.8) is not clear form the results presented here. The authors presented a list of candidates but the lack of a normalization control or validation experiments makes it difficult to identify the true positives (Major comment 5). In addition, the proteomic identification of strep-SecA2 pull-downed fraction provides information on SecA2-binder candidates. It is not clear whether SecA2 participates in the formation of one or more complexes. Although these two pieces of information are descriptive and more-or-less raw results, as long as throughout the text it is not implied that that these are validated results by using strong statements, I think these results still represent an interesting starting point for further analyses on SecA2 function and worth publishing.
The interaction of SecA2 with translating ribosomes is less clear. It is true that ribosomal proteins are detected in SecA2 pull-downs, and that the protein can be partially detected in ribosome co-sedimentation experiments and in polysome-containing fractions. However, the SecA2 can be also detected in non-polysomal fractions, and in the supernatant of co-sedimentation experiments, which makes it difficult to interpret the results. In the absence of negative controls, these results are not solid enough to claim a specific interaction of SecA2 with translating ribosomes (Major comments 1 and 2). Therefore, I would recommend removing the statement from the title (“SecA2 associates with actively translating ribosomes”) and the word “actively” from the abstract (“Here, we demonstrate that SecA2 co-sediments with actively translating ribosomes”). Also, to de-empahsize any other strong statement throughout the paper about this subject. I think the overall story of the paper is not compromised.
As a minor suggestion to improve the readability of the paper, I think the introduction might be amended by including the scientific question and the information necessary to understand why it is important (Minor comment 1).
Author Response
We thank the reviewer for the constructive comments.
The amended version of the manuscript takes into consideration the recommendations made by the reviewer.
- The introduction is amended in the revised version. It provides e.g. information to explain why RNA secretion is important for intracellular bacteria.
- As suggested by the reviewer, we removed word “actively” from the title and the rest of the manuscript with regard to “translating ribosomes”.